# A cationic motif upstream Engrailed2 homeodomain controls cell internalization through selective interaction with heparan sulfates

Sébastien Cardon[1,5], Yadira P. Hervis [1,5], Gérard Bolbach[1,2], Chrystel Lopin-Bon[3], Jean-Claude Jacquinet[3], Françoise Illien[1], Astrid Walrant [1], Delphine Ravault[1], Bingwei He[1], Laura Molina[1], Fabienne Burlina[1], Olivier Lequin [1], Alain Joliot[4], Ludovic Carlier [1]✉ & Sandrine Sagan [1]✉

Engrailed2 (En2) is a transcription factor that transfers from cell to cell through unconventional pathways. The poorly understood internalization mechanism of this cationic protein is proposed to require an initial interaction with cell-surface glycosaminoglycans (GAGs). To decipher the role of GAGs in En2 internalization, we have quantified the entry of its homeodomain region in model cells that differ in their content in cell-surface GAGs. The binding specificity to GAGs and the influence of this interaction on the structure and dynamics of En2 was also investigated at the amino acid level. Our results show that a high-affinity GAG-binding sequence (RKPKKKNPNKEDKRPR), upstream of the homeodomain, controls En2 internalization through selective interactions with highly-sulfated heparan sulfate GAGs. Our data underline the functional importance of the intrinsically disordered basic region upstream of En2 internalization domain, and demonstrate the critical role of GAGs as an entry gate, finely tuning homeoprotein capacity to internalize into cells.

Homeoproteins (HPs) are a large family of transcription factors found in all eukaryotes, which are active during development and in adulthood. Beside regulating gene expression, HPs also act as paracrine factors thanks to their unique ability to travel from cell to cell through unconventional transfer[1]. Their paracrine action implies a direct access of the traveling proteins to the cytosol and nucleus of recipient cells. HP atypical paracrine activity relies on common structural features shared by all HPs[2], and includes specific secretion and internalization motifs[3]. These motifs reside in the 60-residue DNA-binding homeodomain (HD) that defines the HP family (Fig. 1a, b). HDs are organized as three stable helices while the *N*- and *C*-terminal ends are variable and mostly unfolded[4]. The third helix is responsible for the internalization of the protein while the secretion property requires a motif spanning both second and third helices. Notable is that the internalization motif identified 30 years ago within the drosophila *Antennapedia* HD corresponds to the cationic hexadecapeptide Penetratin (RQIKIWFQNRRMKWKK), which has led to the emergence of the field of cell-penetrating peptides (CPPs)[5–7]. Importantly, two modes of internalization co-exist for HPs and CPPs, endocytosis and direct translocation, distinguished by their sensitivity to low temperature[1,5,8,9], and by their ability to direct the protein into the cytosol[10].

[1]Sorbonne Université, École Normale Supérieure, PSL University, CNRS, Laboratoire des Biomolécules (LBM), 75005 Paris, France. [2]Sorbonne Université, Mass Spectrometry Sciences Sorbonne University, MS3U platform, 75005 Paris, France. [3]Univ. Orléans, CNRS, ICOA, 45067 Orléans, France. [4]INSERM U932, Institut Curie Centre de Recherche, PSL Research University, Paris, France. [5]These authors contributed equally: Sébastien Cardon, Yadira P. Hervis. ✉e-mail: Ludovic.Carlier@sorbonne-universite.fr; Sandrine.Sagan@sorbonne-universite.fr

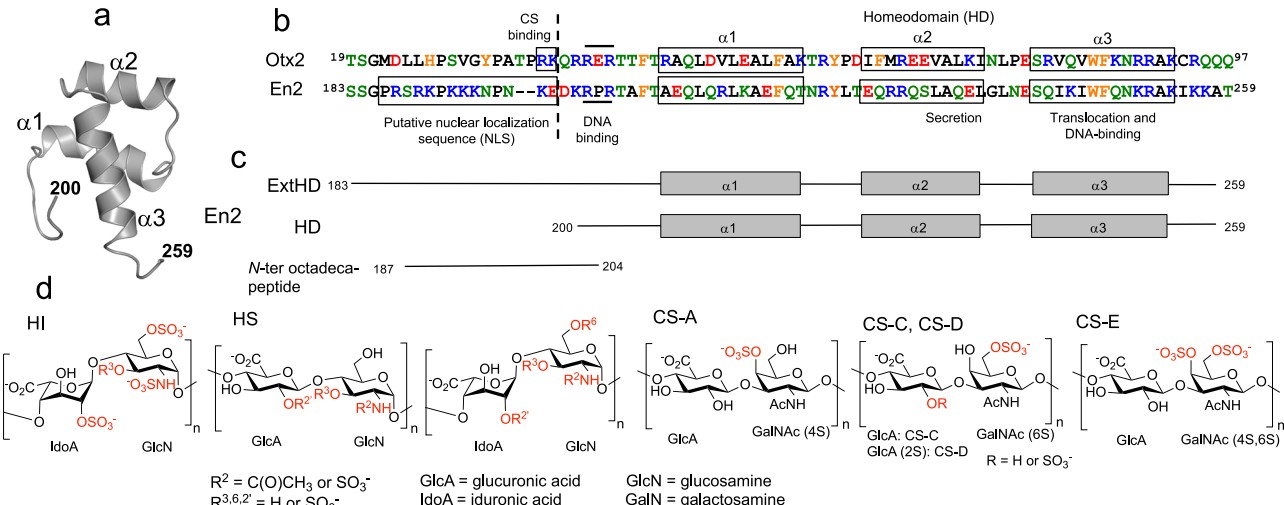

**Fig. 1 | Structure and recognition sequences of En2 region 183-259. a** Solution structure of the homeodomain of *chicken* En2[4] (region 200-259, PDB code 3ZOB). **b** Sequence alignment of En2 region 183-259 (*chicken* gene numbering) with Otx2 region 19-97 (numbering is identical for *chicken* and mammals). Note that these regions are strongly conserved (99% sequence identity between *chicken* and *human* En2, and 100% sequence identity between *chicken* and *human* Otx2). Positively charged, negatively charged, polar, apolar, and aromatic amino acids are colored in blue, red, green, black, and orange, respectively. The putative nuclear localization sequence (NLS) of En2 and the RK doublet of Otx2 that was shown to be crucial for the binding to highly sulfated CS[22] are highlighted by rectangles. **c** Topology of En2 proteins and peptide used in this study. **d** Structure of representative disaccharide units of heparan sulfates (HS), chondroitin sulfates (CS), and heparin (HI).

CPP internalization depends on the presence of cell-surface glycosaminoglycans (GAGs) both in vitro[8,11–14] and in vivo[15]. GAGs are complex polysaccharides that are part of proteoglycans found on the cell-surface and in the extracellular matrix. GAGs bind various extracellular ligands and types of proteins at the cell-surface, which modulate their activity. The functional role of GAGs includes numerous biological processes such as embryonic development, neuronal plasticity, regulation of enzymatic activities, or regulation of cell signaling[16–19]. Regarding HPs, the internalization of extracellular Orthodenticle homeobox 2 protein (Otx2) is restricted in vivo to parvalbumin neurons of the developing mouse visual cortex. Secreted Otx2 accumulates in these cells through interactions with surrounding perineural nets (PNNs) that are enriched in disulfated chondroitin sulfate (CS) GAGs[20,21]. A RK doublet within Otx2 (Fig. 1b), upstream of the highly conserved homeodomain, is responsible for the specific interaction with 2,6 (CS-D) and 4,6 (CS-E) disulfated CS, leading to the restricted internalization of the protein in parvalbumin neurons[22,23].

Another homeoprotein, Engrailed2 (En2), behaves differently. En2 controls the patterning of vertebrate embryos, in particular through the regulation of boundary formation in the developing brain[24,25]. Extracellular En2 poorly accumulates in parvalbumin neurons[22] and the region preceding the homeodomain strongly differs from that of Otx2 (Fig. 1b). In contrast to Otx2, this region is highly enriched in basic amino acids and has sequence similarities with nuclear localization signals (NLS).

In this context, we have questioned herein the influence of GAGs, heparan sulfate (HS) and CS, in the internalization, structure, and dynamics of En2. To decipher the role of En2 interactions with GAGs, we have compared the internalization efficacy of two En2 fragments in model cell lines differing in their cell-surface content in CS and HS. Two protein constructs were produced: the homeodomain alone (HD), containing the minimal cell-penetration Penetratin-like sequence, or extended at its *N*-terminal end with the cluster of basic aminoacids (ExtHD) (Fig. 1c). We have characterized at the molecular level the thermodynamics of interaction of both En2 fragments with selected HS and CS molecules, and studied at the amino-acid level, the impact of these interactions on the secondary and tertiary structure of the proteins. We have identified a high-affinity GAG-binding motif located upstream of the homeodomain, which controls the internalization of En2 through selective interactions with highly sulfated HS at the cell surface.

## Results

### ExtHD internalization depends on cell surface HS

First, the uptake efficacies of both En2 fragments were analyzed, using an adapted MALDI-TOF-based internalization assay[26,27], in two ovarian cell lines that differ in their content of cell-surface glycosaminoglycans. These CHO cell lines present the huge advantage over cultured brain cells that are a mixture of neurons, inter-neurons and glial cells with potential inter-individual variations, of being homogenous in terms of cell phenotype and GAGs content. CHO-K1 cells contain HS and mono-sulfated CS (in a ratio about 50:50)[28,29]. The modified cell line CHO-pgsA-745 (GAG$^{deficient}$) has a genetic deficiency in xylosyl transferase and expresses 10- to 30-fold less HS and CS compared to the parent K1 cell line[28,29]. In K1 cells, HD internalized at very low levels compared to ExtHD (Fig. 2a). After one-hour incubation with each protein added at 7 μM, intracellular HD concentration reached 0.13 μM (assuming a total volume of 1 μL for 10[6] cells). In contrast, ExtHD was internalized 10-fold better. In GAG-deficient cells, internalization of ExtHD was decreased by sixfold, while HD internalization was only decreased by twofold. These results demonstrate that the presence of the *N*-terminal basic extension in ExtHD significantly increases the efficacy of internalization through a process that depends on the presence of cell-surface GAGs. To further analyze the respective contributions of HS and CS on ExtHD internalization, we used different mixtures of enzymes to selectively deplete cell-surface glycosaminoglycans in K1 cells.

Internalization of ExtHD was slightly sensitive to chondroitinase ABC (ChABC) treatment (Fig. 2b). By contrast, heparinase II (HepII) treatment decreased protein internalization by 50%. Combining HepII and ChABC treatments further decreased ExtHD internalization only by 10%. Finally, when cells were treated with mixed heparinases I–III, ExtHD internalization efficacy dramatically dropped to 15% compared to untreated cells, down to levels similar to the ones measured in GAG-deficient cells.

Altogether, the presence of the 17-amino acid extension rich in basic residues that precedes the homeodomain strongly stimulates ExtHD internalization provided that HS are present at the cell-surface,

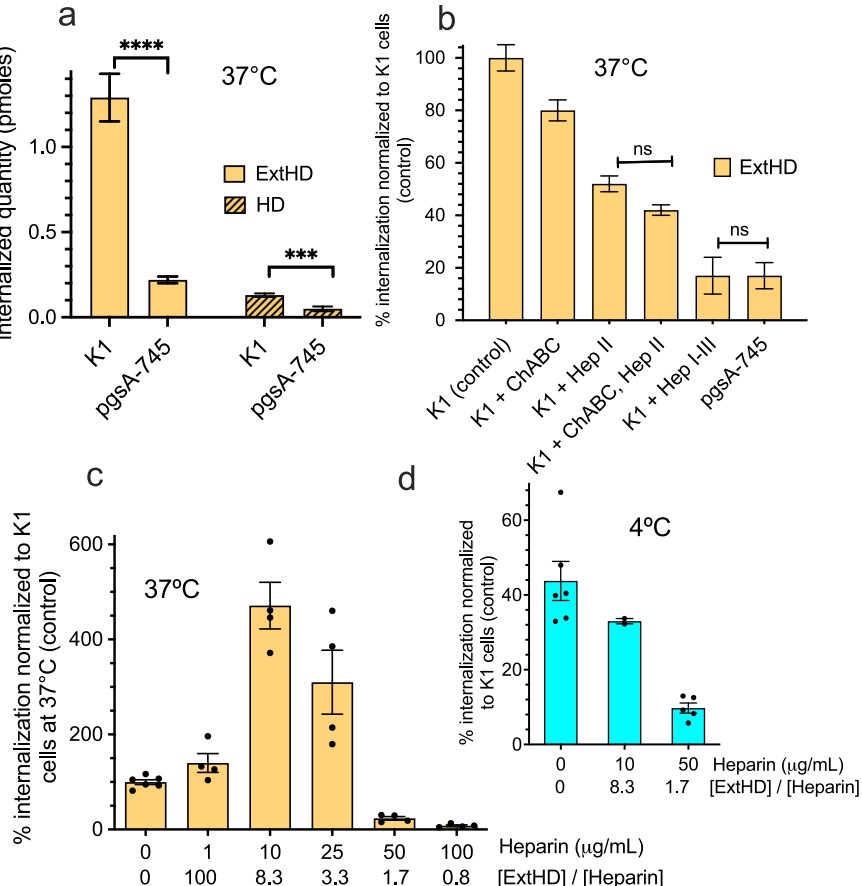

**Fig. 2 | Quantitation of ExtHD and HD internalization in cells and impact of GAGs content. a** Quantity (pmoles) of internalized ExtHD and HD incubated at 7 μM with 10^6 CHO-K1 or GAG^deficient pgsA-745 cells for 1 h at 37 °C (n = 27 and 18 for CHO-K1 and pgsA-745 cells, respectively). CHO-K1 express similar levels of HS and mono-sulfated chondroitins[68]; pgsA-745 express 10–30-fold lower GAGs levels than CHO-K1[28]. **b** Quantity of internalized ExtHD incubated at 7 μM with cells for 1 h at 37 °C (n = 12–18). Data were normalized to the quantity of internalized ExtHD in control CHO-K1 cells. CHO-K1 cells were treated with enzymes degrading HS (heparinases I, II, III) or CS (chondroitinase ABC). **c, d** Quantity of internalized ExtHD in CHO-K1 cells in the absence and presence of increasing amounts of extracellular heparin (n = 2-6). One million cells were incubated with 7 μM protein in DMEM-F12 medium for 1 hr either at 37 °C (**c**) or 4 °C (**d**). Data were normalized to the intracellular quantity of ExtHD in control CHO-K1 cells at 37 °C in the absence of heparin. The ExtHD/heparin molar ratio is indicated for each heparin concentration. Data are presented as mean values +/- SEM. One-way ANOVA and unpaired t tests were used in **a, b** to determine if differences between values were significant: ****$P < 0.0001$, ***$P = 0.0005$, not significant (ns). The list of $P$ values is provided in Supplementary Table 3.

while the GAG-dependence of HD internalization appears less pronounced. The importance of the 17-amino acid extension for ExtHD internalization was confirmed with the SH-SY5Y neuronal cell line that contains similar content of cell-surface HS and CS[30]. As in CHO-K1, ExtHD indeed internalized ~4-fold better than HD in SH-SY5Y cells, and its internalization decreased ~3-fold with heparinase treatment of the cells (Supplementary Fig. 16). It should be noted that although our MALDI-TOF-based internalization assay could not be applied to the full-length En2 protein (see details in the Methods section), the ability of this protein to internalize into CHO-K1 cells was however evidenced qualitatively by fluorescence imaging obtained by confocal microscopy (carboxyfluorescein-labeled full-length En2 is visualized inside cells in images of Supplementary Fig. 19a, b).

### ExtHD recognizes only highly sulfated HS motifs
At the cell surface, HS is a heterogeneous linear polymer that contains highly sulfated regions (so-called S-domains) interspersed with less sulfated and non-sulfated ones[31,32]. To identify if specific structures within HS were responsible for protein binding, we performed a microarray analysis to probe the interaction of ExtHD with 52 structurally defined HS oligosaccharides of different lengths (from 5-mer to 18-mer) and sulfation patterns (Supplementary Fig. 1). As shown in Supplementary Figs. 2 and 3, the oligosaccharides that bind the more

efficiently ExtHD are the 12-mer compounds **18** and **19**, which have the greatest number of sulfates and the highest charge densities (i.e., number of sulfates per disaccharide unit).

Notable is the presence of two fully sulfated disaccharide units (*N*-, 2-*O*, 3-*O* and 6-*O* sulfation) in compound **18** and one in compound **19**. A significant fluorescent signal was also observed for the 12-mer compound **22** that contains 4 trisulfated disaccharide units lacking 3-*O* sulfation. By contrast, the 12-mer compounds containing only one or two sulfate groups per disaccharide unit (compounds **23** to **25**) did not yield significant signals, regardless of sulfate distribution. Investigating oligosaccharide length, only low-intensity signals were observed for the sulfated compounds of the 6-, 7- and 9-mer series, including the highly sulfated compound **16** (6-mer, 8 sulfates), indicating that ExtHD better recognizes longer HS fragments (≥12 saccharide residues), which is supported by the significant fluorescence signal recorded for compound **27** (Supplementary Fig. 3). Overall, this screening profile shows that ExtHD binds selectively to long and highly charged HS, with a strong preference for trisulfated and tetrasulfated disaccharide units.

### Soluble HI discriminates the two internalization modes of ExtHD
To further characterize the role of HS in ExtHD internalization, the effect of exogenous heparin addition was analyzed on ExtHD internalization. The pharmaceutical heparin used in this study is a highly

**Table 1 | Interaction thermodynamics of the *N*-ter octadecapeptide and proteins with heparin (12 kDa, net charge about −55), studied by ITC**

| protein (z+, z−) | NaCl (mM) | $K_d$ (nM) | $\Delta H$ (kJ/mol) | $-T\Delta S$ (kJ/mol) | $n$ (prot/poly-sacch.) | Complex net charge |
|---|---|---|---|---|---|---|
| Full length En2 (44 +, 31-) | 100 | 570 ± 360 | -51 ± 8 | 10 ± 9 | 2.9 ± 0.4 | 39 +/55- |
| *N*-ter octadecapeptide (10 +, 2-) | 100 | 300 ± 67 | −81 ± 7 | 44 ± 5 | 7 ± 1 | 56 +/55- |
| HD (14 +, 6-) | 0 | 870 ± 120 | −86 ± 10 | 51 ± 10 | 4.3 ± 0.5 | 38 +/55- |
| | 100 | 2090 ± 10 | −46 ± 8 | 14 ± 7 | 5.3 ± 0.5 | |
| | 200 | nd | nd | nd | nd | |
| ExtHD (21 +, 7-) | 0 | 50 ± 10 | −91 ± 2 | 50 ± 8 | 3.5 ± 0.5 | 45 +/55- |
| | 100 | 120 ± 18 | −41 ± 2 | 1 ± 1 | 2.9 ± 0.6 | |
| | 200 | 620 ± 130 | −19 ± 2 | −16 ± 2 | 3.4 ± 0.1 | |

The proteins were titrated with the polysaccharide at 25 °C in 50 mM NaH₂PO₄, pH 7.4. Data are the mean ± SD ($n = 3$).
*nd* not determined.

sulfated polymer of ~12 kDa, mostly composed of trisulfated GlcNS,6S−IdoA2S which is found in *N*-sulfated domains of HS[33,34] and is reported as a key structure for protein recognition[35]. Heparin exhibited bimodal concentration-dependent effects at 37 °C, either promoting or inhibiting ExtHD internalization (Fig. 2c). At concentrations lower than 25 µg/mL (protein/heparin molar ratio >3), heparin increased intracellular ExtHD up to 5-fold while concentrations above 50 µg/mL (protein/heparin molar ratio <2) resulted in a substantial decrease of ExtHD internalization, almost to complete inhibition at 100 µg/mL. Interestingly, analysis by dynamic light scattering (DLS) of the mixed heparin/ExtHD solutions (Supplementary Fig. 4) revealed the formation of aggregated species (up to 40%) of µm size at low heparin concentration (10 µg/mL), but neither with the protein alone nor when heparin and ExtHD were in stoichiometric quantities. In contrast, we did not observe aggregated species with the heparin fragment dp20 (Supplementary Fig. 4) and concomitantly did not measure any increase in ExtHD internalization in the presence of dp20, in contrast to heparin (Supplementary Fig. 18). Since the formation of GAG/protein aggregates was previously shown to promote endocytosis of Penetratin-like CPPs[13,14,36], we analyzed the effect of decreasing temperature on ExtHD internalization efficacy in CHO-K1 cells.

At low temperature (4 °C), endocytosis pathways are totally inhibited, while direct crossing of the lipid bilayer, which is not strictly energy-dependent, still occurs[8,37,38]. In the absence of exogenous heparin, ExtHD internalized in CHO-K1 at 4 °C with 44 ± 6% efficacy compared to 37 °C control condition (Fig. 2d), indicating that the protein efficiently internalizes through direct translocation in cells. Contrasting with 37 °C conditions, at 4 °C the presence of 10 µg/mL heparin in the culture medium resulted in the inhibition of ExtHD internalization in K1 cells (Fig. 2d). Thus, low concentration of soluble heparin can fulfill, even outperform, the requirement of membrane-associated forms of HS for ExtHD endocytosis likely through formation of large complexes between heparin and ExtHD. Conversely, translocation would specifically require interaction with membrane-associated HS, which is antagonized by soluble heparin. Corroborating these results, ExtHD translocation (at 4 °C) was impaired in GAG-deficient CHO psgA-745 cells that have reduced levels of membrane-bound HS (Supplementary Fig. 17).

### The high binding affinity of ExtHD for HI relies on the *N*-ter peptide
To further understand the molecular basis of En2 recognition by highly sulfated HS, we used ITC to determine the thermodynamic parameters of heparin interaction with HD and ExtHD proteins (Table 1 and Supplementary Fig. 5). The two protein fragments differed in their affinities for heparin, ExtHD exhibiting a 20-fold greater affinity than HD (Table 1 and Supplementary Fig. 5). These interactions were enthalpy-driven, showing that electrostatic interactions are involved in the

formation of the complex between either protein and heparin, as supported by the strong dependence of binding affinity on salt concentration. Interestingly, the synthetic octadecapeptide corresponding to the cationic region that precedes the helical core of En2 homeodomain (aa 187–204; Fig. 1c) had similar affinity for heparin as the entire ExtHD protein, indicating that this motif is the main determinant of HS interaction within ExtHD. The ExtHD fragment can accommodate 6–7 disaccharides (20 disaccharides in heparin / 3 bound proteins) while HD and the *N*-ter octadecapeptide can bind 4 (20/5) and 3 (20/7) disaccharides, respectively. As the stoichiometry (polysaccharide/protein or 1/n determined in ITC experiments) of interaction between ExtHD and heparin corresponds to the sum of the stoichiometry of interaction measured with the *N*-ter octadecapeptide and HD alone respectively, cooperative events in this interaction appear unlikely. Altogether, these results indicate that the *N*-ter octadecapeptide confers a unique property to the ExtHD protein in terms of binding to highly sulfated HS. It should be noted that the full-length En2 protein interacts with heparin (Table 1 and Supplementary Fig. 6) with an affinity about ~5-fold lower than that of ExtHD, the stoichiometry being identical for the two proteins (3 proteins bound per HI polymer). This indicates the absence of additional heparin binding site in the full-length En2 protein. The lower affinity of full-length En2 for heparin arises from an unfavorable entropy contribution, likely because of the presence of the large *N*-terminal disordered region that partly compensates the strong favorable enthalpy contribution (see discussion in caption of Supplementary Fig. 7).

### Similar binding affinity of ExtHD and HD for phospholipids
Since protein-lipid interactions play a key role in membrane-translocation properties of homeodomains[10], we next investigated whether enhanced affinity with phospholipids also contribute to the higher internalization efficacy of ExtHD. In particular, we previously showed that the interaction of En2 homeodomain with anionic lipid vesicles induces a major rearrangement of its tertiary structure[4]. Herein we used ITC to probe the interaction of HD and ExtHD with 100 nm large unilamellar vesicles (LUVs) that we prepared either from 1-palmitoyl-2-oleoyl-*sn*-glycero-3-phospho-(1′-rac-glycerol) (POPG), 1-palmitoyl-2-oleoyl-*sn*-glycero-3-phosphocholine (POPC), or a combination of both. The presence of one saturated and one unsaturated fatty acyl chains makes these phospholipids good mimics of mammalian cell membrane composition.

With pure zwitterionic lipids (Table 2), no interaction occurred with either of the two proteins used at 20 µM as previously reported for the HD construct[4]. Addition of anionic POPG to POPC to obtain LUVs with 50/50 POPG:POPC and 70/30 POPG:POPC composition, had no effect. Significant ExtHD and HD interactions were observed only with pure POPG LUVs, showing that negatively charged partners are mandatory for interaction of the proteins with lipid membranes. The

dissociation constant value determined for POPG was the same for the two proteins (≈ 7 μM; Table 2). Thus, the presence of the cationic N-terminal extension does not strengthen the interaction of ExtHD with anionic phospholipids.

### ExtHD contains two independent GAG-binding sites

We next investigated the molecular basis of the interaction between En2 and highly sulfated GAGs at the single residue level using hetero-nuclear single-quantum correlation (HSQC) NMR experiments. To identify GAG-binding regions of ExtHD, we performed chemical shift perturbation (CSP) experiments by collecting $^1H$-$^{15}N$ HSQC spectra on uniformly $^{15}N$-labeled samples in the presence of increasing amounts of unlabeled oligosaccharides. Our initial attempts to titrate $^{15}N$-labeled ExtHD with 12 kDa heparin resulted in dramatic line broadening of NMR signals (Supplementary Fig. 9), in agreement with the formation of a high molecular weight complex as determined by ITC (3:1 protein:GAG complex of 40.5 kDa). Since both the glycan microarray analysis and ITC indicated that ExtHD can accommodate oligosaccharide fragments containing ~6–7 trisulfated disaccharides, we selected a heparin dodecasaccharide (dp12) fragment for NMR titrations. As shown in Fig. 3a, the addition of stoichiometric amounts of heparin dp12 induced large chemical shift changes in the $^1H$-$^{15}N$ HSQC of ExtHD, consistent with a fast exchange regime on the chemical shift time scale and a dissociation constant in the micromolar-millimolar range. An increase in the line width of the NMR cross-peaks

was only observed for a few residues, indicating that the binding to the heparin fragment does not induce oligomerization of the globular homeodomain.

The highest perturbations were mostly localized in the cationic region $^{189}$RKPKKKNPNKEDKRPR$^{204}$, which includes the basic residues preceding helix α1 known to interact with DNA (Fig. 3b). Large CSP values were also observed at the C-terminal tail of the homeodomain that encompasses three positively charged residues ($^{254}$KIKKAT$^{259}$). By contrast, residues of the helical domain displayed rather modest chemical shift changes in the presence of 1.0 molar equivalent of heparin dp12. Additional large perturbations are nevertheless observed beyond a 1:2 protein:GAG ratio in helix α1 (K216 and F219), the loop α1-α2 (R223 and T226), and helix α3 (Q249, N250, K251, and R252). Apparent thermodynamic affinities ($K_d^{app}$) were determined at the residue level from saturation curves (Fig. 3c). Variable $K_d^{app}$ values in the low micromolar range (~4–10 μM) were obtained in the N-terminal region, the highest binding affinities being observed in the basic epitope 189-204 (Supplementary Table 2). The C-terminal residues 256-258 exhibit $K_d^{app}$ values in the range of 70-130 μM and bound heparin dp12 with a significantly lower affinity. In the case of residues R223 and T226, the binding curves were characteristic of very weak interactions with $K_d^{app}$ values in the millimolar range. As a result, the loop α1-α2 did not contribute significantly to the binding to heparin despite the large CSP values measured in this region above a protein:GAG ratio of 1:2.

Taken together, our data reveal that the helical core of En2 homeodomain is flanked by two heparin-binding motifs with residues 192–204 forming the major binding site, and the C-terminal residues 254-259 forming an additional, low-affinity binding region. To confirm that these two motifs do not bind heparin in a synergistic manner, we titrated heparin dp12 at increasing concentrations into a solution of $^{15}N$-labeled HD. The absence of residues 183–199 in the HD construct does not modify significantly the perturbation profile of residues 200–259, which CSP values were remarkably similar to the ones measured with the longer ExtHD construct (Supplementary Fig. 10).

### Table 2 | Thermodynamics of interaction of ExtHD and HD with POPG LUVs (100 nm), determined by ITC

| Protein | $K_d^{apparent}$ (μM) | $\Delta H$ (kJ/mol) | $-T\Delta S$ (kJ/mol) | $n$ (lipids/protein) |
|---|---|---|---|---|
| HD | 7.5 ± 2.5 | 1.6 ± 0.1 | −30 ± 5 | 7.6 ± 0.8 |
| ExtHD | 6.8 ± 2.3 | 3.1 ± 1.3 | −29 ± 3 | 8.7 ± 2.3 |

The proteins were titrated by LUVs at 25 °C in 50 mM NaH$_2$PO$_4$ buffer, pH 7.5. Data are the mean ± SD (n = 3).

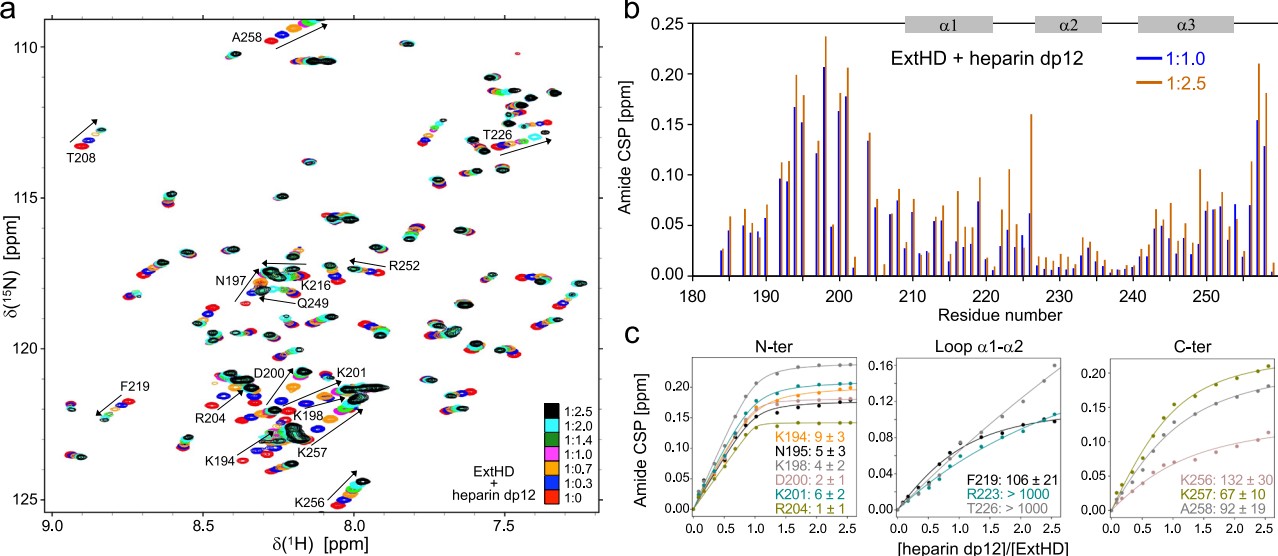

**Fig. 3 | NMR characterization of ExtHD interaction with heparin dp12 at the amino acid level.** Interaction of ExtHD with heparin dp12 is probed in 50 mM NaH$_2$PO$_4$ (pH 7.4) and 100 mM NaCl. **a** Overlay of $^1H$-$^{15}N$ HSQC spectra (500 MHz) obtained for $^{15}N$-ExtHD in the absence and presence of increasing amounts of heparin dp12. Stoichiometry between ExtHD and heparin dp12 is: 1:0 (red), 1:0.3 (dark blue), 1:0.7 (orange), 1:1 (pink), 1:1.4 (green), 1:2 (light blue) and 1:2.5 (black).

Amino acid residues exhibiting the highest perturbations (arrows) are indicated in the spectra. **b** Amide CSP of ExtHD induced by the addition of 1.0 and 2.5 molar equivalents of heparin dp12. **c** Saturation curves corresponding to the binding to heparin dp12 are shown for residues displaying the highest perturbations with experimental data represented by points and non-linear curve fitting by lines. $K_d^{app}$ values are given in micromolar.

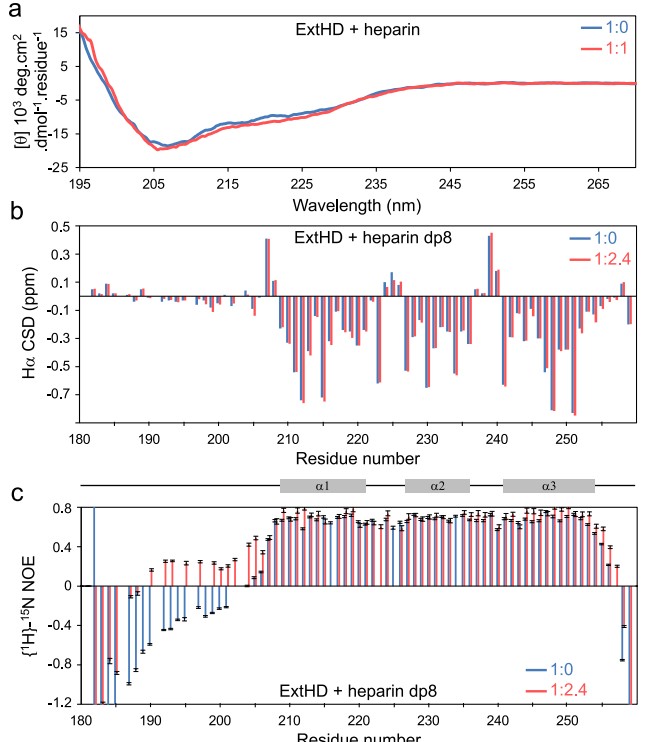

**Fig. 4 | The GAG-binding region 189-204 of Engrailed2 remains disordered upon binding to heparin or heparin fragments. a** CD spectra of ExtHD in the absence (blue) and presence (red) of 1 molar equivalent 12 kDa heparin. Chemical shift deviations (CSD) and {¹H}-¹⁵N NOEs of ExtHD recorded at 500 MHz in the absence (blue) and presence (red) of 2.4 molar equivalents heparin dp8 are shown in **b, c**, respectively. Error bars in **c** indicate the standard error for each value based on peak intensity relative to background noise (see the "Methods" section).

This observation is consistent with the existence of independent interacting regions that bind heparin with differential affinities.

To identify basic residues within the *N*-terminal region that predominantly contribute to heparin binding, we produced three ExtHD mutants in which three basic pairs in the region ¹⁸⁹RKPKKKNPNKEDKRPR²⁰⁴ were individually replaced by serine pairs. ITC data revealed that all three mutants lose affinity for heparin compared to the parent protein, from 5-fold for the R189S/K190S mutant up to 8.5-fold for the K193S/K194S one (Supplementary Table 1 and Supplementary Fig. 5). Since the two mutants K193S/K194S and K201S/R202S exhibited the lowest binding affinities, in the μM range, the basic residues within the region 192–204 are mostly responsible for the binding to heparin.

### The *N*-ter GAG-binding motif bound to HI fragments remains disordered

We next investigated the influence of heparin binding on the conformational properties of En2 region 183-259. For that purpose, we examined the site-specific secondary structure and dynamics of ExtHD in the absence and presence of heparin fragments using ¹Hα chemical shift deviations (CSDs) and heteronuclear ¹H-¹⁵N NOEs. The stability of the NMR samples containing stoichiometric amounts of heparin dp12 was not long enough to record these experiments. Therefore, we used the shorter heparin dp8 fragment, which provided more stable samples and binds ExtHD with a similar CSP profile (Supplementary Fig. 10) albeit with a slightly weaker affinity (Supplementary Table 2). CSDs are defined as the differences between measured chemical shifts and corresponding random coil values for each residue[39]. Previous studies have shown that residues 183–206 upstream of the homeodomain are highly disordered in an

En2 fragment encompassing residues 146–259[40,41]. Accordingly, small near-zero values characteristic of disorder were observed in the region 183–206 of ExtHD in the absence of heparin dp8 (Fig. 4b). By contrast, the three α-helices of the homeodomain were well identified by three continuous stretches of negative CSDs. The addition of 2.4 molar equivalents of heparin dp8 did not induce significant changes in the ¹Hα chemical shifts of ExtHD, and the CSD diagram of the bound form was virtually identical to that of the free form. Together with the fact that the presence of heparin fragments only induced selective perturbations of the ¹H-¹⁵N chemical shifts, our data indicate that the binding to heparin dp8 does not modify the secondary and tertiary structure of ExtHD. To verify that the conformational properties of ExtHD do not depend on oligosaccharide chain length, we recorded circular dichroism (CD) spectra in the absence and presence of 1 molar equivalent of 12 kDa heparin. The presence of heparin did not modify significantly the CD spectrum (Fig. 4a), confirming that ExtHD, and in particular the *N*-terminal residues 192-204 do not undergo major structural rearrangement upon GAG binding. We measured heteronuclear {¹H}-¹⁵N nuclear Overhauser effects (NOEs) to probe backbone motions of ExtHD that occur on the picosecond-to-nanosecond timescale (Fig. 4c). In the free form, the disordered nature of the *N*- and *C*-terminal residues flanking the helical core of the homeodomain was confirmed by negative NOEs that are indicative of high flexibility. Upon binding to heparin dp8, NOE values were found to increase in the *N*-terminal region 190-205 and at the *C*-terminal extremity, evidencing that the interaction with GAGs significantly restricts internal motions in these regions. However, the amplitude of NOEs in the bound form remained weak in the *N*-terminal region (<0.25) when compared to the rigid helical core that exhibits NOEs >0.7. As a result, the backbone of the *N*-terminal GAG-binding site retained substantial mobility when bound to heparin, which supports a binding mode mostly driven by electrostatic contacts.

### ExtHD selectively interacts with HI compared to disulfated CS

As evidenced above, ExtHD preferentially interacts with highly sulfated regions of heparan sulfates. Since other GAG types can harbor high sulfate content, such as sulfated chondroitin (CS) enriched in the PNNs of developing cortex, we investigated whether ExtHD also recognizes structurally defined CS oligosaccharides. The NMR titrations of ExtHD with CS-A and CS-C dp6 fragments did not result in significant modifications of protein chemical shifts. By contrast, the CSP pattern induced by CS-E dp6 showed strong similarities with that obtained with heparin dp6 (Supplementary Fig. 11). Large CSP were observed in both the *N*- and *C*-terminal regions flanking the helical domain and the loop α1-α2. Similar to what was observed with heparin fragments, the greatest apparent binding affinities were obtained in the *N*-terminal region 192-204 (Supplementary Fig. 11 and Supplementary Table 2). These data suggest that heparin and CS-E oligosaccharides share similar ExtHD-binding sites. However, in comparison with heparin dp6, the CSP magnitude and apparent binding affinities are weaker in both the *N*-terminal region and the *C*-terminal extremity.

To confirm the lower affinity of ExtHD for CS-E, we used ITC and titrated HD and ExtHD proteins with a commercially available CS-E fragment of 72-kDa (Supplementary Fig. 8). Because the size of this polysaccharide differs from that of heparin, i.e. 12 kDa (~20 disaccharides) for heparin and 72 kDa (~135 disaccharides) for CS-E, the binding affinity for each oligosaccharide type was determined by dividing the measured free energy by the number of bound protein molecules per polysaccharide determined by ITC (see Supplementary Information for further explanation). Our results indicate that ExtHD displays a 126-fold greater affinity for heparin than for CS-E (Table 3). In addition, ExtHD has 17-fold better affinity than HD for heparin whereas the two proteins have similar affinity for CS-E. These data confirm that the presence of the cationic *N*-terminal extension confers to the

**Table 3 | Interaction thermodynamics of the *N*-ter octadeca-peptide and proteins with heparin and CS-E, studied by ITC**

| Protein | Polysacch. | $\Delta G_n$ (kJ/mol) | *n* protein/ polysacch. | $\Delta G_n/n$ (kJ/mol) | $K_d$ mM |
|---|---|---|---|---|---|
| *N*-ter | heparin | −37 | 7 | −5.3 | 117 |
|  | CS-E | −43 | 55 | −0.78 | 730 |
| HD | heparin | −32 | 5 | −6.4 | 75 |
|  | CS-E | −44 | 33 | −1.3 | 591 |
| ExtHD | heparin | −40 | 3 | −13.3 | 4.5 |
|  | CS-E | −41 | 29 | −1.4 | 568 |

The proteins were titrated with the polysaccharides at 25 °C in 50 mM NaH$_2$PO$_4$, 100 mM NaCl, pH 7.4. The affinity of peptide or protein for the polysaccharide is determined by dividing the free Gibbs energy by *n* (stoichiometry of the complexes) and calculated as $K_d = e^{\Delta G/RT}$. Data are presented as the mean (*n* = 3).

homeodomain a higher selectivity of recognition for trisulfated HS (heparin over CS-E: selectivity of 126-fold for ExtHD versus 8-fold for HD construct).

## Discussion

The internalization pathways of homeoproteins and of CPPs derived from the third helix of homeodomains still need to be fully characterized[7,42]. Similarly to the requirement of disulfated GAGs for Otx2 internalization demonstrated in vivo, internalization of Penetratin (the highly conserved third α-helical hexadecapeptide of homeodomains) depends on the presence of cell-surface GAGs[8,12–15,43,44]. Therefore, one could assume that Penetratin-like CPPs, HDs, and HPs share at least some steps of their mechanisms of internalization[45]. Using a combination of quantitative methods, we explored the role of specific cell-surface GAGs in the recognition and internalization process of the homeoprotein Engrailed2 (En2), and intended to identify protein motifs involved in the interaction.

Our data show that the ExtHD fragment encompassing En2 residues 183–259 enters cells principally through interaction with glycosaminoglycans of heparan sulfate (HS) type, thanks to the presence of a unique GAG-binding motif [189]RKPKKKNPNKEDKRPR[204] that selectively binds highly sulfated HS. Truncation of this high affinity GAG-binding motif that precedes the helical core of the homeodomain drastically reduces its internalization in the CHO cell model. NMR studies reveal that the *C*-terminal extremity of the homeodomain, also interacts with heparin, however with about two orders of magnitude less affinity. As demonstrated by ITC and NMR data, the two main GAG-binding regions flanking the helical domain interact with heparin independently from each other and not synergistically.

The high-affinity HS-binding sequence lies in a highly flexible region of the protein[40,41], which remains disordered upon binding to GAGs. It contains the BBXBX motif (where B is a basic residue) previously described as a consensus HS-binding sequence[46]. It also holds the KKK sequence identified as one of the most significant tripeptides (99th percentile) found among 437 heparin-binding proteins[47]. Accordingly, mutation of basic pairs within the [192]KKK[194] and [201]KRPRT[205] sequences lead to a significant reduction in the binding affinity for heparin. Mutation of the basic pair R189/K190 also decreases the affinity for heparin, showing that the full hexadecapeptide region 189-204 is required to maintain high-affinity binding to heparin. We can thus assume that the full HS binding motif of En2 corresponds to the sequence (B)$_2$X(B)$_3$(X)$_3$BAABBXXB where A is an acidic residue (B and X being respectively a basic and any AA). Interestingly, this motif corresponds to an evolutionary conserved region in engrailed gene sequences.

An important feature of cell-surface HS is the high variability in size, composition, and sulfate distribution of the sulfated regions (S-domains) in which disaccharide units can be sulfated at 4 distinct

positions. In addition, the disaccharide composition of HS is known to vary from one cell type to the other[35,48]. Using a glycan microarray analysis, we showed that En2 region 183–259 binds preferentially to long (>dp12) and highly sulfated HS oligosaccharides. The correlation between binding affinity and HS sulfate density, together with the strong dependence of the interaction on salt concentration supports a charge-based binding mode. Accordingly, the high-affinity HS-binding motif of En2 can also accommodate the highly sulfated polysaccharide CS-E, albeit with a lower affinity in comparison with the binding to heparin. In addition, the presence of soluble CS-E in the extracellular medium was shown to either promote or inhibit En2 internalization in K1 cells, depending on the GAG concentration, as observed with soluble heparin (Supplementary Fig. 18). These data suggest that membrane surface CS-E could also play a role in En2 internalization.

A specific feature of the internalization of HP and HP-derived domains (HD, Penetratin) is the co-existence of two mechanisms, endocytosis and translocation. For all, internalization indeed occurs at 4 °C while endocytosis is prevented, and mutations that decrease Penetratin uptake efficiency, such as the replacement of the critical Trp residue by Phe, also prevent HD and HP internalization[5,49]. Translocation implies a transient perturbation/disorganization of the lipid bilayer[3,37,45,50] and it has been recently shown in cell culture that direct interactions between En2 and the anionic phospholipid PIP2 promote the translocation of the protein towards the cytosol[10]. Moreover, Carlier et al. demonstrated that interaction of En2 homeodomain with anionic phospholipids is required to induce a conformational change in its 3D structure, leading to the insertion of the third helix within the acyl chains of the lipid bilayer[4]. The plasma membrane is a highly asymmetric structure. In particular, negatively charged lipids are under-represented in the outer membrane leaflet facing the extracellular space, raising the issue of the initiation of the translocation event leading to internalization.

Our data herein support a role of HS as a critical actor of En2 internalization not only through endocytosis[51], but also through direct translocation across the plasma membrane. En2 translocation (measured at 4 °C) is reduced in GAG-deficient cells compared to wild-type cells. Based on previous reports[4] and the data in the present work, we propose that sulfated GAGs anchored by proteins at the cell surface would be good candidates as translocation initiators (Fig. 5). In this model, the highly sulfated S-domains of HS attract En2 at the cell-surface through electrostatic interactions with the HS-binding motif that precedes the homeodomain. The dynamic nature of these interactions allows a fast diffusion of En2 within the extracellular matrix leading to its stepwise accumulation at the close vicinity of the plasma membrane. Indeed, the selective inhibitory action of low concentrations of soluble heparin on translocation but not on endocytosis, supports the need for a proximal distance between GAGs and the lipid bilayer to evoke translocation. This initial interaction with HS would bring the homeodomain in close contact to the acyl part of the lipid bilayer[52], leading to the modification of its conformation and the insertion of the third helix within the lipid bilayer. The interaction of En2 with PIP2 in the inner leaflet, shown to regulate the bidirectional exchanges of the protein between the cytosol and the lipid bilayer[10], would eventually result in the transfer of the protein within the cytosol. Following this model, we propose that the bidirectional transfer of En2 is regulated by a dynamic equilibrium between protein-lipid or protein-GAG interactions depending on the side of the plasma membrane. It is interesting to note that our DLS and NMR data do not support the oligomerization of ExtHD when bound to heparin fragments from dp4 to dp20. Therefore, En2 oligomerization does not seem to be a prerequisite for membrane translocation, as already proposed for its secretion[53].

The interaction of GAGs with extracellular HPs has important consequences at the physiological level, as the paracrine action of HPs relies on their ability to be internalized. For instance, the plasticity of

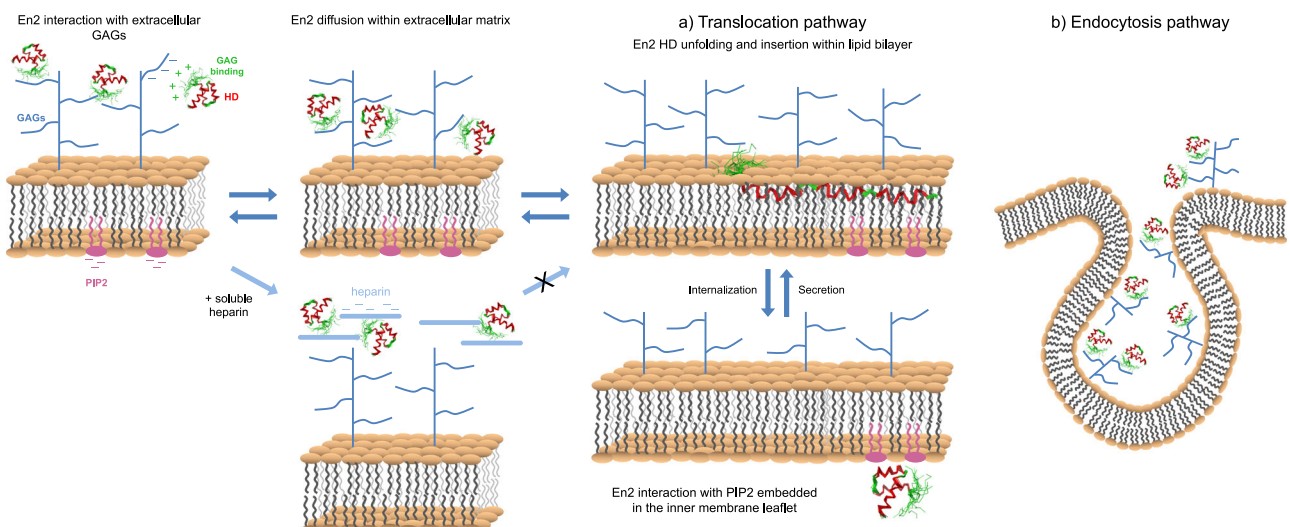

**Fig. 5 | Hypothetical model of GAG-dependent bidirectional transfer of HPs across the plasma membrane.** The role of highly sulfated HS in cell internalization of En2 through translocation and endocytosis pathways is illustrated in panels **a** and **b**, respectively.

the visual cortex relies on the ability of secreted Otx2 to internalize into parvalbumin neurons through interactions with the surrounding extracellular matrix enriched in disulfated GAGs[22,23]. It is rather interesting however to mention that the molecular composition and glycan structure of CSPGs vary in PNNs[54]. This heterogeneity of PNNs is crucial since it impacts the localization of the homeoprotein Otx2, thus controls the functional maturation of PV cells, but it makes it more challenging to define a CS-E structure representative of that found in PNNs. The axonal guidance of retinal neurons also depends on the paracrine action of HPs, Vax1 at the optic chiasma and later, En2 in the tectum[55,56]. Vax1 internalization requires the presence of HS on retinal neurons, and for En2, its delivery into the cytosol is required to exert its paracrine action that involves the translational activation of Ephrin-mediated axon guidance[55,57]. Since a graded distribution of En2 in the extracellular matrix of the tectum is necessary for the proper positioning of retinal axons in the developing brain[58], GAGs might also act to restrict HP diffusion in the extracellular space[53,59], protecting them from enzymatic degradation (Supplementary Fig. 12).

A critical component of HP paracrine action is the restriction of their internalization to specific cell types. Due to their widely heterogenous structures, GAGs are attractive candidates to modulate the cellular tropism of HP internalization. Compared to En2, the binding specificity for GAGs is markedly different in the case of Otx2 homeoprotein, which contains only 5 positively charged residues, including a RK doublet, in the N-terminal extension preceding the homeodomain helical core (Fig. 1). In vivo, an Otx2 fragment encompassing the homeodomain and the RK doublet is selectively internalized into cortical neurons surrounded by perineuronal nets similarly to the full protein, while the corresponding AA double-mutant is not. Mutation of the RK doublet by alanines in an Otx2 pentadecapeptide ([36]RKQRRERTTFTRAQL[50]) fully abolishes peptide binding to CS-D and CS-E oligosaccharides[22]. These results show that the RK doublet upstream of the homeodomain is responsible for the specific interaction of Otx2 with cell-surface CS and its subsequent internalization in PNN-positive cells. Interestingly, the GAG-interacting domains within En2 and Otx2 strongly differ, the RK doublet that recognized disulfated CS in Otx2 being replaced by a KE doublet in En2, which is frequently found in HS-binding proteins[47]. In addition, En2 region 183-259 (ExtHD) interacts more efficiently with heparin than disulfated CS-E, while Otx2 displays higher affinity for CS-E and CS-D as compared to heparin[22]. Accordingly, En2 is significantly less internalized in PNN-positive cells than Otx2[22]. Overall, these findings support the idea that the cationic motifs preceding the homeodomain region control the spatial

distribution of extracellular homeoproteins through a fine tuning of their binding specificity for GAGs. This underlines the importance of these GAG-binding motifs that define a functional region involved in HP trafficking, acting in concert with internalization/secretion motifs.

## Methods

### Expression and purification of recombinant proteins

Regions 200-259 (HD) and 183-259 (ExtHD) of *chicken* Engrailed2 (Uniprot accession number Q05917) as well as the full-length protein (residues 1-289) were expressed as (His)6-tagged Cherry fusion proteins using pSCherry1 vectors and the *E. coli* SE1 strain as expression host (Eurogentec, Seraing, Belgium). ExtHD mutants K189S/K190S, K193S/K194S, and K201S/R202S were expressed as (His)$_6$-tagged GST fusion proteins using pGEX-6P plasmids (Genscript, Piscataway, NJ, USA) and the E. coli BL21-CodonPlus-RIL expression strain (Agilent Technologies, Santa Clara, CA, USA). A PreScission protease cleavage site (LEVLFQ/GP) was introduced prior to the En2 sequence in all constructs. As a result of PreScission cleavage, final proteins have four extra residues at the N-terminus (GPAS). For internalization assays, an additional Cys residue was introduced at the N-terminus by site-directed mutagenesis (leading to GPCAS). Unlabeled fusion proteins were produced from cells cultured in LB rich medium. Uniformly labeled $^{15}N$ and $^{15}N,^{13}C$ proteins were overexpressed in M9 minimum media using the procedure developed by Marley et al. for preparing high yields of uniformly labeled proteins[60]. Protein expression was induced by adding 1 mM isopropyl-1-thio-β-D-galactopyranoside (IPTG) to the cell cultures followed by a 4-h incubation period at 37 °C under agitation. Cells were pelleted by centrifugation (3000 × g for 20 min), resuspended in a 20 mM NaH$_2$PO$_4$ buffer (pH 7.4) containing 500 mM NaCl, 30 mM imidazole and 1 mM DTT, and lysed by sonication. Soluble fractions containing En2 proteins were separated from cell debris by centrifugation (30,000 × g for 40 min at 4 °C), and loaded on a Ni-NTA column (GE Healthcare, Chicago, IL, USA). Proteins were purified using IMAC standard protocols with a 30–250 mM imidazole gradient. Eluted fractions were pooled and dialyzed against PBS buffer, and (His)$_6$-tagged PreScission protease was added to cleave fusion proteins. A second step of purification was further performed using Ni-NTA chromatography to remove the cleaved (His)$_6$-tagged fusion partner and the (His)6-tagged PreScission protease. Purity of protein fractions was controlled by SDS-PAGE analysis, and protein identity was checked by MALDI-TOF analysis. Fractions containing pure target proteins were finally concentrated by ultrafiltration using a 3-kDa cutoff membrane (Millipore). Protein folding was controlled by

NMR. In particular, the $^1$H,$^{15}$N chemical shifts of the homeodomain region in the E.coli-expressed ExtHD (Fig. 3a) and HD (Supplementary Fig. 10) proteins were very similar to those previously reported for the native helix-turn-helix fold of En2 homeodomain[4], indicating that the region of the homeodomain adopts its native fold in our E.coli-expressed proteins. The synthetic N-ter octadecapeptide used in ITC experiments (biotin sulfone-G4-RSRKPKKKNPNKEDKRPR-CONH$_2$) was synthesized by the Boc strategy, purified by RP-HPLC (purity >95%) and checked by MALDI-TOF mass spectrometry (expected mass: 2744.5 Da; measured mass: 2744.9 Da).

## Preparation of $^{14}$N, $^{15}$N biotin-labeled recombinant proteins

Biotin-labeled proteins were used in internalization assays to fish the internalized species out the cell lysates through specific binding to streptavidin-coated magnetic beads. The two recombinant proteins expressed in *E. coli* contain an additional Cys on the N-terminal end. Purified proteins were transferred to a buffer containing 50 mM sodium phosphate (pH 7.0), 150 mM NaCl, 10 mM EDTA, and 400 μM TCEP. The protein solution was degassed for 30 min under vacuum before addition of 5 equivalents of maleimide-PEG2-biotin (MW = 525) (Sigma Aldrich, Saint-Louis, MO, USA). The reaction was carried out under inert atmosphere (N$_2$) to prevent oxidation of the maleimide-PEG2-biotin and decrease of the reaction yield. The reaction solution was protected from the light and stirred for 90 min at room temperature. The reaction was then repeated with addition of fresh maleimide-PEG2-biotin solution. Proteins were transferred into a buffer containing 20 mM NaH$_2$PO$_4$ (pH 7.2), 20 mM NaCl, and stored at −80 °C. The yield of protein biotinylation was evaluated to be >90% for HD and ExtHD, obtained from MALDI-TOF MS analyses (Supplementary Fig. 14). In the case of the full-length En2 protein, we encountered many difficulties in obtaining the biotin-labeled protein in a sufficient yield (> 50%) to perform the quantification assays, even when trying different strategies. In addition, we could not recover the intact protein after incubation with cells, likely because the large N-terminal disordered region is unstable and partially subject to in-cell proteolysis during the assays. Other attempts to introduce one biotin to the full-length protein, either to the native Cys178 or by addition of an extra Cys at the C-terminal end of the protein were also unsuccessful: the yield of biotinylation was too low (~20%) in the first case and biotin was found inaccessible to streptavidin-coated beads, in the second one.

## Cell culture

Wild type Chinese Hamster Ovary CHO-K1 (https://www.atcc.org/products/ccl-61), GAG-deficient CHO-pgsA745 cells (GAG-deficient, https://www.atcc.org/products/crl-2242) and SH-SY5Y (https://www.atcc.org/products/crl-2266) (ATCC, LGC Standards S.a.r.l. - France) were cultured in DMEM-F12 medium (Life Technologies, Carlsbad, CA, USA) supplemented with 10% fetal calf serum (FCS), penicillin (100,000 IU/L), streptomycin (100,000 IU/L), and amphotericin B (1 mg/L) in a humidified atmosphere containing 5% CO2 at 37 °C. SH-SY5Y cells were cultured in DMEM-F12 medium supplemented with 10% FCS and differentiated by a sequential treatment with retinoic acid (RA) and brain derived neurotrophic factor (BDNF). The protocol of differentiation consisted in the addition of 10 μM RA (Sigma Aldrich, Saint-Louis, MO, USA) in DMEM-F12 containing 15% FCS, 2 days after seeding. Five days later, cells were washed three times with DMEM-F12 and incubated with 50 ng/mL BDNF (Sigma Aldrich, Saint-Louis, MO, USA) in serum-free medium for 24 h.

## Enzymatic removal of glycosaminoglycans from cell surface

Cells were washed with PBS and incubated with enzymes, alone or in mixture: heparinase I, heparinase II, heparinase III or/and chondroitinase ABC (Sigma Aldrich, Saint-Louis, MO, USA) at a final concentration at 0.1 U/mL in PBS pH 7.4 (containing calcium and

magnesium) for 1 h at 37 °C. Cells were then washed with HBSS, then with DMEM F12 before further incubation with the proteins.

## Quantification of protein internalization by mass spectrometry

A MALDI-TOF-based internalization assay was previously shown to accurately quantify the cellular uptake of biotinylated cell-penetrating peptides[26]. We have adapted this protocol to proteins (Supplementary Fig. 13). The method relies on the calibration of the internalized unlabeled protein with the addition of a known amount of its isotopically $^{15}$N-labeled counterpart just prior cell lysis and before affinity capture through a biotin bait introduced on a cysteinyl residue added at the N-terminus (Supplementary Fig. 14). The quantification is achieved through the ratio of the respective integrated peak area of the non-labeled and $^{15}$N labeled protein, here used as internal standard. The rate of labeling was calculated from the ion signals (m/z) to be >90%. Despite this high rate of labeling, the $^{14}$N and $^{15}$N protein ions have overlapped m/z signals in MALDI-TOF MS spectra. To overcome this overlapping, we developed a software allowing the deconvolution of the signals and an accurate fit of the ratio between the $^{14}$N non-labeled and $^{15}$N labeled species (see below for details). By comparing the experimental mass spectra to those obtained from the mixing of the individual spectra respectively for the labeled and the non-labeled protein, we were able to determine the absolute quantity of protein internalized inside cells (Supplementary Fig. 15).

Briefly, adherent and confluent cells ($10^6$ cells/well in 12-well plates) were incubated with 500 μL of a $^{14}$N-protein in culture medium at 7 μM for 60 min at 37 °C. After washing, trypsin treatment of cells permits to digest membrane-bound protein species and to detach cells. Cells are lysed with a hypertonic detergent solution containing a known amount of the $^{15}$N-labeled proteins and boiled. After centrifugation to remove cell debris, the lysate is incubated with streptavidin-coated magnetic beads for 1 h to extract $^{14}$N- and $^{15}$N-proteins. After several washing steps, proteins are eluted from beads in 3 μL of CHCA matrix at room temperature for 15 min. We found CHCA the most suitable matrix for elution, incorporation in the matrix crystals, and MALDI-TOF analysis. We used purified CHCA and classical calibrant mixtures, such as Peptide Mix3 and Protein Mix2, from LaserBio Labs (Sofia Antipolis, France). In all MALDI-TOF experiments, we worked with saturated CHCA solution in 1:1 ACN:H$_2$O (0.1% TFA). The samples are then analyzed by MALDI-TOF MS (positive ion linear mode) on a Voyager-DE Pro mass spectrometer (Applied Biosystems, Foster City, CA, USA). Experiments were first done to determine the accurate amount of $^{15}$N-protein to be added to obtain as close as possible a 1:1 ratio between the internalized $^{14}$N and $^{15}$N internal standard proteins. This first step allows one to get accurate, robust and reproducible MS quantification results[61]. Each experiment was done in triplicate and independently repeated at least 3 times.

## Software developed to quantify the cellular uptake of proteins by mass spectrometry

In a previous study[26], MALDI-TOF was found to be well adapted to quantify the cellular uptake of biotinylated cell-penetrating peptides. This method was based on the addition of a known and appropriate amount of the labeled peptide ($^2$H deuterium labeling) to the cell lysis buffer before extraction of the two peptide species ($^2$H-labeled and non-labeled). As in conventional internal standard experiment the quantification is achieved through integrating the peaks area of both labeled and non-labeled peptides. For proteins with higher mass than peptides, as those studied herein, we have developed a software allowing an adjustment of the ratio r between non-labeled ($^{14}$N)/labeled ($^{15}$N) proteins, by comparing the experimental mass spectra profile to that obtained from the mixture of separate labeled and non-labeled protein (Supplementary software file). The validity of such an approach originates from the properties of both proteins: same sequence implying identical ionization yield, similar incorporation in

the MALDI matrix, same peak broadening due to attachment of alkaline and matrix adducts, identical detection efficiency due to the small mass difference. In addition, if experimental mass spectra of $^{14}$N, $^{15}$N and the mixture ($^{14}$N and $^{15}$N) from internalization are obtained under the same experimental conditions (delayed ion extraction, ion detection and laser fluence), it is possible after mass calibration to calculate a theoretical mass spectrum for a mixture with a ratio r and to compare it to the experimental mass spectrum. The software calculates, normalizes and displays the experimental and simulated mass spectra and a fast adjustment of the ratio *r* can be obtained. Baseline correction on the calculated mass spectrum can be also implemented to get a better adjustment simulated vs experimental. Studying the different charge states allows to estimate the accuracy of the ratio.

The non-complete labeling of the protein can be also studied by comparison of $^{14}$N and $^{15}$N mass spectra through the calculated convolution of the $^{14}$N ion peak (+1, +2, +3…) with a Gaussian distribution of $^{15}$N atoms by adjusting the mean value <15 N > and the standard deviation ($\sigma$). The convolution operates over the m/z range in which the $^{14}$N peak is defined. The display of the convoluted and $^{15}$N experimental peak allows one to deduce the value of <15N> and $\sigma$. If needed, experimental data can be integrated before processing the convolution.

Experimental MS spectra are extracted as ASCII files (m/z, intensity) from the processing software (Data Explorer) of the MALDI-TOF apparatus (DE-Pro and 4700 Proteomics Analyzer both from Applied Biosystems, Foster City, CA, USA). The software to determine the ratio *r* and also the labeling distribution, if any, has been written in Visual Basic (V 6.0). The simplest case is that of a complete labeling for which non-labeled and labeled peaks are superimposable and thus the *m/z* shift of the non-labeled peak overlay that of labeled peak using an intensity ratio (Supplementary Fig. 15). To take into account a labeling distribution, each experimental point of the unlabeled peak is shifted in m/z with a relative intensity modulated by the probability of the shift. A new peak is formed after adding all the contribution for a value m/z within a chosen window (1-5 Da depending on the protein mass).

In Supplementary Fig. 15, an example is shown with the ExtHD protein as unlabeled ($^{14}$N) and labeled ($^{15}$N) versions. No $^{15}$N distribution was necessary to fit the +1 and +2 protein peaks showing that the labeling was complete. The experimental data of $^{15}$N were well fitted from the $^{14}$N data shifted by 128 Da for both +1 (lower mass spectrum) and +2 (upper mass spectrum) charge states.

## Microarray of structurally defined HS compounds
A custom HS-microarray was designed by the company Glycan Therapeutics (https://www.glycantherapeutics.com/services/microarray-analysis), as described[62]. 52 structurally defined HS oligosaccharides were synthesized and immobilized on a microarray chip. The biotinylated ExtHD protein was incubated at 100 nM with the microarray slide for 1 hr in a Tris/phosphate buffer pH 7.50 containing 137 mM NaCl and 10% bovine serum albumin (BSA)[62]. After washing with the Tris/phosphate buffer, the microarray slide was treated with 10 μg/mL Alexa-Fluor 488-labeled Avidin. The wash process was repeated before scanning fluorescence of the array slide at 488 nm.

## Isothermal titration analysis (ITC)
Experiments were done with a nano-ITC calorimeter (TA Instruments, New Castle, DE, USA). Measurements were done at 25 °C in 50 mM NaH$_2$PO$_4$ pH 7.4 and various NaCl concentrations (0, 100, 200 mM). Titration of the proteins in the ITC cell was done by 25 successive injections of 10 μL of polysaccharide (heparin, heparin-derived, CS-E, obtained from Iduron, Cheshire, UK) with 5 min injection intervals. Proteins, peptide and polysaccharides were used in different concentrations to get accurate fitted curves (between 20-80 μM for

proteins and peptide and 40-120 μM for GAG). Control experiments were obtained by injections of buffer into proteins and injections of GAGs into buffer. Data were analyzed with NanoAnalyze software provided by TA Instruments.

LUVs were prepared as described[63]. Briefly, lipid films were prepared by dissolving phospholipids into chloroform or a mixture of chloroform and methanol (2/1, vol/vol). Formation of the lipid film was achieved by evaporating solvents under nitrogen flux, then drying in a vacuum chamber for at least 1 hr. Films were then hydrated with 10 mM sodium phosphate buffer pH 7.5 and extensively vortexed at a temperature superior to the lipid phase transition temperature to obtain MLVs. To form LUVs, the MLVs were subjected to five freeze/thawing cycles. The homogeneous lipid suspension was then passed 19 times through a mini-extruder (Avanti Polar Lipids, Avanti Alabaster, AL, USA) equipped with two stacked 100 nm polycarbonate membranes at a temperature above the phase transition temperature of the phospholipid. Titrations were performed by injecting aliquots of LUVs (1 mg/mL) into the calorimeter cell containing the protein solution (0.1 mM), with 5 min intervals between injections.

## Circular dichroism (CD)
CD spectra were measured on a Jasco J-815 spectropolarimeter (JASCO Corporation, Tokyo, Japan) over the wavelength range 190–270 nm, by using a 0.1-cm path-length quartz cell (internal volume 200 μL) from Hellma (Muellheim, Germany). Measurements were carried out at 25 °C with a 1 nm/min scan speed and a band width of 1 nm. Proteins and oligosaccharides were dissolved in a buffer containing 20 mM sodium phosphate (pH 7.4) and 100 mM NaF. Four scans were accumulated and averaged for each sample. All spectra were corrected by subtraction of the background obtained for each protein-free mixture. Protein concentration was 10 μM in the absence and presence of 1 molar equivalent heparin.

## NMR data collection and analysis
NMR experiments were acquired at 298 K on a Bruker Avance III 500 MHz spectrometer equipped with a 5-mm TCI cryoprobe. NMR spectra were processed and analyzed with the software programs NMRPIPE[64] and NMRFAM-SPARKY[65]. All experiments were performed with NMR samples containing $^{15}$N or $^{15}$N,$^{13}$C uniformly labeled HD or ExtHD, previously dissolved in a buffer containing 40 mM sodium phosphate (pH 6.3), 100 mM NaCl, and 7.5% D$_2$O. 0.1 mM Sodium 2,2-dimethyl-2-silapentane-5-sulfonate-d6 (DSS) was added in the NMR samples as an internal 1H chemical shift reference. $^{13}$C and $^{15}$N chemical shifts were referenced indirectly to DSS, using the absolute frequency ratios. Sequence-specific assignments of backbone $^{15}$NH, $^{1}$HN, $^{13}$Cα, $^{1}$Hα, $^{13}$CO, and side-chain $^{13}$Cβ, $^{1}$Hβ resonances were obtained from the analysis of a series of three-dimensional triple resonance experiments (HNCA, HNCO, HN(CA)CO, HNCACB, CBCA(CO)NH, HNHA, and HBHA(CO)NH), recorded on samples containing 600 μM $^{15}$N,$^{13}$C ExtHD. Chemical shift perturbation experiments (CSP) were performed by collecting 1H-$^{15}$N HSQC spectra on $^{15}$N uniformly labeled HD or ExtHD (100-150 μM) in the absence and presence of increasing amounts of heparin fragments (obtained from Iduron, Cheshire, UK) or structurally defined CS-E dp6 that was chemically synthesized[66], previously dissolved in the NMR buffer. CSP values were calculated for each residue from the differences in 1HN ($\triangle\delta_{H^N}$) and $^{15}$N ($\triangle\delta_N$) chemical shifts between the free and the bound states using the relation: $CSP = \sqrt{\left(\Delta\delta_{H^N}\right)^2 + \left(0.1\Delta\delta_N\right)^2}$. Apparent thermodynamic affinities (Kd$^{app}$) at the residue level were calculated for residues in the fast exchange regime from a nonlinear least squares curve fitting of CSPs upon oligosaccharide addition using in-house Python scripts[67]. Steady state $^{15}$N-{$^{1}$H} NOE values in the absence and presence of heparin dp8 (Iduron, Cheshire, UK) were determined as the ratio of peak heights in

$^{15}$N-$^1$H spectra collected with and without proton saturation during the recycle delay. $^1$H saturation was achieved by applying a train of 120° pulses spaced at 5 ms interval for a period of 4.5 s. Uncertainty in the $^{15}$N-{$^1$H} NOE values were calculated from the root mean square baseline noise of both spectra estimated with NMRPIPE.

## Reporting summary
Further information on research design is available in the Nature Portfolio Reporting Summary linked to this article.

## Data availability
Additional data that support this study are available from the corresponding authors upon request. Source data are provided with this paper.

## Code availability
The software developed to quantify the cellular uptake of proteins by mass spectrometry is available in the Supplementary Software file and requires Visual Basic to run it.

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

## Acknowledgements

This research has received fundings from the Agence Nationale de la Recherche (grants No. ANR-BLAN2016-CROSS, A.J., S.S., L.C., A.W., F.B., B.H., F.I., D.R., Y.P.H., O.L., S.C., C.L.-B.; ANR-20-CE44-0018-GLYCO-TARGET, S.S., A.W., F.I., D.R., L.C., O.L., Y.P.H., C.L.-B., A.J.).

## Author contributions

L.C. and S.S. designed research; S.C., G.B., Y.P.H., C.L.-B., J.-C.J., F.I., A.W., D.R., B.H., L.M., A.J., L.C., and S.S. performed research; S.C., G.B., Y.P.H., F.B., O.L., L.C., and S.S. analyzed data. G.B. C.L.-B. and J.-C.J. contributed reagents/analytical tools; L.C. and S.S. wrote the manuscript.

## Competing interests

The authors declare no competing interests.
