## [Peer Review File · Nature Communications]

REVIEWER COMMENTS

Reviewer #1 (Remarks to the Author):

Dear Editor,

Please, find my comments regarding the reviewing of the manuscript from Cardon et al. (ref NCOMMS-21-44879), entitled “A cationic motif in Engrailed-2 homeoprotein controls cell internalization through selective interaction with heparan sulfates”.

In this manuscript, the authors investigated in detail the interaction of homeoprotein transcription factor Engrailed-2 (En2) with complex polysaccharides of the Glycosaminoglycan (GAG) family, using a panel of biochemical and biophysical techniques. Results demonstrate binding of both Heparan sulfate (HS) and chondroitin sulfate (CS)-E to a hexadecapeptide GAG-binding motif, and that interaction with cell-surface HS is required for efficient cell internalization.

Despite some concerns detailed below, my overall assessment is that this study is well conducted, data are sound and results provide very interesting insights into a dual role of HS polysaccharides in Eng2 internalization. Beyond the field of homeoproteins, GAG driven internalization mechanisms still remain poorly understood, which makes this study fully relevant. As a GAG expert, I consider that these results will be of high interest to the field.

Main scientific comment:

1- GAGs are a main focus of the manuscript, they ought to be better introduced. Please add 1-2 sentences in the introduction for presenting GAGs as complex, sulfated polysaccharides with wide protein binding/modulating properties, and a couple of recent reviews about GAG structure and function could also be cited to provide the reader with up to date general information about these polysaccharides.

2- Page 3: PNN feature very specific GAG content and are, as stated by the authors, notably enriched in disulfated chondroitin sulfate. Cited refs 16 and 17 do not actually support this particular statement. Please add adequate reference.

3- Page 3: “with disulfated chondroitin with sulfation at position 6 (CS-E, CS-D)” : the authors should more precisely define CS-D and CS-E as 2,6 and 4,6 disulfated CS, respectively.

4- Page 5 : Structure of HS : “highly sulfated regions (so-called S-domains) interspersed with less sulfated and nonsulfated regions”: please add a review reference.

5- Page 5-6: microarray binding analysis from figure provides very interesting data and I overall agree with the authors conclusions. However, I find intriguing that the best binding oligosaccharides (18 and 19) feature rare 3S groups and that the short 3-sulfated heptasaccharide 4 also shows signal above background.

3-O-sulfation has been recently involved in the binding and internalization of Tau (Zhao et al., *Angew Chem Int Ed Engl* 2020 Jan 27;59(5):1818-1827 and others), which could be relevant for the present study. This paper in particular used the same oligosaccharide microarray as in the present study. In that case, HS 3O-sulfated AT-III binding motif resulting from Hs3st1 activity was identified as the binding motif, whereas in the present study HSV gD-binding motif (IdoA2S, GlcNS3S,6S : only present in 18) could be preferentially involved in the binding to ExtHD, which would explain the major signal difference observed between 18 and 19 (only one sulfate difference between the two oligosaccharides).

CHO cells do not express (or express very little) Hs3sts. Therefore, it would be highly interesting to assess if overexpression of one HS3st (HS3st5 would be the best candidate) would enhance further EXTND internalization.

6- Page 12: “indicating that N-sulfation of the amino sugar is not mandatory for the binding to ExtHD”. The fact that GalNAc containing CS-E and heparin oligosaccharides binds similarly to ExtHD does not necessarily mean to me that GlcNS residues of heparin are not involved. Spatial distribution of sulfates may be different in HS and CS.

7- Page 13: the authors claims that affinity of ExtHD (and HD) is greater for Heparin than for CS-E. I would be more cautious on this conclusion. Whereas Heparin is mostly composed of IdoA2S, GlcNS6S disaccharide, the proportion of CS-E units in a CS-E polysaccharide can vary a lot according to the polysaccharide origin. A disaccharide analysis of both heparin and CS-E would be needed to clarify this. Besides charge content, distribution of E units along the chain (in blocks or dispersed) may also be critical. It is therefore difficult to know whether the CS-E used in this study is structurally representative of that found in a relevant tissue/cell surface (or PNN). In line with this, analysis of the interaction of ExtHD with synthetic CS-E dp6 (exclusively composed of E units) by NMR show fairly similar binding (figure S9 and table S2). This should be discussed in the manuscript.

8- I find the dual role of HS in the internalization of En2 through endocytosis and direct translocation across the plasma membrane very interesting. Surprisingly, the authors chose to depict only the translocation mechanism in their figure 5 model. I understand that this process is their main focus, but I

feel that the figure should gain by including both internalization mechanisms. Clatrin dependent HSPG mediated endocytosis has been reported in previous studies. Did the authors try to inhibit this pathway in their 37°C internalization assay?

9- In figure 2A-B, the authors show the role of HS in EXT₂ internalization using CHO mutant cells or heparinase/chondroitinase digestions. However, these experiments do not provide any information about a potential role of CS-E. As stated by the authors, CHO cells do not produce CS-E. To address this, the internalization assay would have to be performed on CS-E producing cells. Alternatively, could soluble CS-E be used in competition, as in figure 2C-D? It would also be interesting to know whether CS-E could mediate both endocytosis and direct translocation mechanisms.

10- Table I: I do not understand the 80- charges for 12 kDa Heparin. As stated by the authors, 12 kDa Heparin is ~20 disaccharide, and the average charge of heparin is ~2.5-2.8 charges per disaccharide (with variations according to the source of material), which makes an average 50-56 global negative charge. Again, disaccharide analysis of the heparin used would clarify this.

Minor issues

1- Figure 1 : HI not defined. Heparin ? Why HI?

2- The authors used a very elegant MS-based approach to quantify protein internalization. Although the method has been thoroughly described elsewhere, addition of a supplementary figure summarizing the principle of the technique may be considered ?

3- Table 3 : I presume K_d are in nM, not mM

4- page 23 BSA : bovine and not bovin

Reviewer #2 (Remarks to the Author):

In this manuscript the authors discovered that Engrailed-2 (En2) is a novel HS-binding protein and performed a series of biochemical and biophysical experiments to characterize this En2-HS interaction. While the authors did a decent job in characterizing the interaction, their study is incomplete for three reasons as will be detailed below (points 1-3). The biggest problem of this study however is that it is out of biological context. It is paramountly important to use a relevant cell type to investigate the role of HS in En2 internalization, which might be quite different from what they observed in CHO-K1 cells.

1. Based on the NMR titration experiments using dp8 oligosaccharide, the authors concluded En2 contains two GAG-binding sites. I'm not convinced because this experiment used dp8, much shorter than the required minimum binding length (dp12) as suggested by the microarray analysis. It is quite possible that these two sites work in a synergistic manner when exposed to dp12. The evidence presented in Fig.S8 is not sufficient to exclude this possibility. A far better way to reveal whether the two motifs really work together to form a complete HS-binding site would be to truncate/mutate a few basic residues at the C-terminal sequence.

2. There is a high probability that HS might induce EN-2 to form oligomers, which the authors did not investigate. This experiment will need to be done with dp12 to prevent aggregation as suggested by Fig. S4.

3. I don't understand why only truncated En-2 was used in the study. Does full-length En-2 bind HS? If so, how does the affinity compare to ExtHD?

4. Another major concern is that how did the authors verify that the E.coli expressed En-2 is biologically active?

Reviewer #3 (Remarks to the Author):

The manuscript investigates the mechanism of the cellular internalization of the Engrailed-2 (En2), a “mobile” transcription factor that is involved in the brain development of vertebrates, focusing on the crucial role of GAG at the level of the cell membrane. The authors hypothesize that the interaction with highly sulfated GAG guides the internalization of the protein En2. To demonstrate this hypothesis, a multimodal and complementary analysis, based on MALDI mass spectrometry measurements, bindings thermodynamic and conformational changes study, was performed to study the biochemistry of the cellular internalization of the protein. The authors could show how chemically the membrane composition can affect the internalization of the transcription factor En2 that was not well understood previously. The understanding of biochemistry beyond protein internalization is highly interesting. Significant results, obtained successfully by MALDI MS, show that protein internalization yields are driven by the presence of an N-terminal basic extension and different sulfate patterns on the cell membrane. The topic will be relevant to other fields such as oncology since the En2 is involved also in cancerogenesis. I found the manuscript well-structured and written as a non-native English speaker. The multi-modal approach employed by the authors supports the conclusions of the mechanism of internalization of En2.

For this reason, I recommend the publication after addressing my concerns about the introduction, the MALDI MS part of material and methods and supporting information.

I found the introduction well written but I think it is missing the reason why it is important to study the internalization of En2 and why it is important to understand the mechanism to emphasize the importance of the work especially for a Nature journal. I have found several publications regarding the role of En2 not only in neural formation but also in cancer such as prostate cancer (<https://www.nature.com/articles/s41598-019-41678-0>) I think giving other examples of applications can catch the interests of researchers, who are working in different fields.

As an expert in MALDI-TOF MS, I found a very innovative method to study the cellular uptake of proteins especially because they can give quantitative data of the internalized protein. However, I have the following concern about the method:

The authors mentioned that “The validity of such an approach originates from the properties of both proteins: same sequence implying identical ionization yield, similar incorporation in the MALDI matrix, same peak broadening due to attachment of alkaline and matrix adducts, identical detection efficiency

due to the small mass difference". In my experience, differences in ionization efficiency can occur when spotting the same analyte on the target plate is performed manually. How is the sample spotted on the target plate to ensure that the sampling and therefore the incorporation of the protein with the MALDI matrix are the same? I think the authors should specify the matrix application in the material methods and prove the technical reproducibility. The MALDI-MS was performed three times. Is it possible to estimate the coefficient of variation to show the technical reproducibility of the method especially the spot application on the target plate?

The formulation of the matrix solution should be provided (solvent and concentration) in material and methods

Which calibrant was employed for the MALDI MS measurement?

Why the matrix CHCA was chosen for this study? Generally, proteins are well detectable with sinapinic acid as a matrix.

In Figure S5, the x-axis shows injection time and not mole ratio. Is there any reason for this inconsistency?

Figure S12 is very difficult to understand. The first picture is the double charged form of the protein. The second is the single charged form and the third is the comparison between simulated and experimental mixtures. First question: I do not understand why the mass of the double charged is the same for the two isotopes? Moreover, what does mean shift at 128? This picture needs to be more described with more details.

In Figure S13, why the relative internalized quantity of K1 is not the same as the K1 control in Figure 2D?

I would be happy to review a revised manuscript

Reviewer #4 (Remarks to the Author):

This is a fantastic piece of work, dealing with an important topic and solved using a multidisciplinary approach, with a variety of structural and biophysical techniques on top of the molecular biology protocols employed to prepare the target protein. Technically, the work is excellent, the NMR approach is perfect, the quality of the spectra is very high and the discussion and conclusions are clearly stated. Nothing else to say.

Point-by-point response to the reviewers' comments

Reviewer #1 (Remarks to the Author):

Dear Editor,

Please, find my comments regarding the reviewing of the manuscript from Cardon et al. (ref NCOMMS-21-44879), entitled "A cationic motif in Engrailed-2 homeoprotein controls cell internalization through selective interaction with heparan sulfates".

In this manuscript, the authors investigated in detail the interaction of homeoprotein transcription factor Engrailed-2 (En2) with complex polysaccharides of the Glycosaminoglycan (GAG) family, using a panel of biochemical and biophysical techniques. Results demonstrate binding of both Heparan sulfate (HS) and chondroitin sulfate (CS)-E to a hexadecapeptide GAG-binding motif, and that interaction with cell-surface HS is required for efficient cell internalization.

Despite some concerns detailed below, my overall assessment is that this study is well conducted, data are sound and results provide very interesting insights into a dual role of HS polysaccharides in Eng2 internalization. Beyond the field of homeoproteins, GAG driven internalization mechanisms still remain poorly understood, which makes this study fully relevant. As a GAG expert, I consider that these results will be of high interest to the field.

We sincerely thank the reviewer for their overall assessment of the manuscript.

Main scientific comment:

1- GAGs are a main focus of the manuscript, they ought to be better introduced. Please add 1-2 sentences in the introduction for presenting GAGs as complex, sulfated polysaccharides with wide protein binding/modulating properties, and a couple of recent reviews about GAG structure and function could also be cited to provide the reader with up to date general information about these polysaccharides.

We have now added the following couple of sentences in the introduction part as well as recent reviews.

Glycosaminoglycans are complex polysaccharides that are part of proteoglycans found on the cell-surface and extracellular matrix. GAGs bind various extracellular ligands and types of proteins at the cell-surface, which modulate their activity. The functional role of GAGs includes numerous biological processes such as embryonic development, neuronal plasticity, regulation of enzymatic activities, or regulation of cell signaling: 1) Xu and Esko, Demystifying heparan sulfate-protein interactions. *Annu Rev Biochem.* (2014) 83:129-57. doi: 10.1146/annurev-biochem-060713-035314.; 2) Smith et al. "GAG-ing with the neuron": The role of glycosaminoglycan patterning in the central nervous system. *Exp. Neurol.* 2015, 274 (Pt B), 100–114.; 3) Mikami and Kitagawa, Chondroitin Sulfate Glycosaminoglycans Regulate Distinct Cell Surface Receptor-Mediated Neuronal Functions, *Trends in Glycoscience and Glycotechnology* (2021) 33(191), E11-E16, 10.4052/tigg.2004.1E; 4) Vallet, et al. The glycosaminoglycan interactome 2.0. *American journal of physiology* (2022) 322(6), C1271-C1278; 10.1152/ajpcell.00095.2022).

2- Page 3: PNN feature very specific GAG content and are, as stated by the authors, notably

enriched in disulfated chondroitin sulfate. Cited refs 16 and 17 do not actually support this particular statement. Please add adequate reference.

We thank the reviewer for their comment that is totally correct. We have now replaced the previous refs 16 and 17 by the following references: Deepa, et al. Composition of perineuronal net extracellular matrix in rat brain: a different disaccharide composition for the net-associated proteoglycans. *J. Biol. Chem.* 281(26), 17789–17800 (2006). doi:10.1074/jbc.M600544200 ; Carulli, et al. Composition of perineuronal nets in the adult rat cerebellum and the cellular origin of their components. *Comp Neurol.* (2006) Feb 1;494(4):559-77. doi: 10.1002/cne.20822.

3- Page 3: “with disulfated chondroitin with sulfation at position 6 (CS-E, CS-D)”: the authors should more precisely define CS-D and CS-E as 2,6 and 4,6 disulfated CS, respectively.

We thank the reviewer for their comment. The CS-D and CS-E have now been defined precisely in the text p.3 as 2,6- and 4,6-disulfated CS.

4- Page 5: Structure of HS : “highly sulfated regions (so-called S-domains) interspersed with less sulfated and nonsulfated regions”: please add a review reference.

The following reference has now been added to the manuscript: Benito Casu, Chapter 1 - Structure and Active Domains of Heparin, in *Chemistry and Biology of Heparin and Heparan Sulfate* (2005) Pages 1-28. Editor(s): Hari G. Garg, Robert J. Linhardt, Charles A. Hales. <https://doi.org/10.1016/B978-008044859-6/50002-2>.

5- Page 5-6: microarray binding analysis from figure provides very interesting data and I overall agree with the authors conclusions. However, I find intriguing that the best binding oligosaccharides (18 and 19) feature rare 3S groups and that the short 3-sulfated heptasaccharide 4 also shows signal above background.

3-O-sulfation has been recently involved in the binding and internalization of Tau (Zhao et al., *Angew Chem Int Ed Engl* 2020 Jan 27;59(5):1818-1827 and others), which could be relevant for the present study. This paper in particular used the same oligosaccharide microarray as in the present study. In that case, HS 3O-sulfated AT-III binding motif resulting from Hs3st1 activity was identified as the binding motif, whereas in the present study HSV gD-binding motif (IdoA2S, GlcNS3S,6S : only present in 18) could be preferentially involved in the binding to ExtHD, which would explain the major signal difference observed between 18 and 19 (only one sulfate difference between the two oligosaccharides). CHO cells do not express (or express very little) Hs3sts. Therefore, it would be highly interesting to assess if overexpression of one HS3st (HS3st5 would be the best candidate) would enhance further EXTND internalization.

We thank the reviewer for their interesting comment. Specificity of 3-O-sulfation for HS-binding proteins has indeed been reported and was recently investigated using the same microarray approach as in the present study (Horton et al., 2021, Construction of heparan sulfate microarray for investigating the binding of specific saccharide sequences to proteins, *Glycobiology*, 31, 188-199). In this study, antithrombin (AT), which high-affinity interaction with HS requires the rare 3-O-sulfation, was shown to recognize only compounds 4, 16, 18 and 19 of the microarray (see microarray results below provided by GlycanTherapeutics and derived from Horton et al., 2021). All four HS oligosaccharides recognized by AT contain 3-O-sulfation. Their size is very different, ranging from 6 to 12 monosaccharides. The screening profile shown below is remarkably different from that of our ExtHD protein (Fig. S2), which poorly recognizes the 6-mer compound 16 that contains one 3-O sulfate group but binds significantly the 12-mer compound 22 that lacks 3-O sulfation. Furthermore, while oligosaccharides 4, 20, and 27 are secondarily recognized by our ExtHD protein, compound

4 contains 3-O-sulfation but the two others do not. The charge density of compounds 4 and 20 are similar (respectively 8 charges / 7 monosaccharides and 16/12). For compound 27, the charge density is not above the threshold but its number of monosaccharides is the highest (18) in the series. Finally, the analysis of fluorescence intensity versus sulfate density in Fig. S3 supports that the net charge of HS is essential for ExtHD binding. From the comparison with AT protein, we are convinced that the strong binding of ExtHD to the highly-charged oligosaccharides 18 and 19 containing the 3-O-sulfation arises from an increased charge density rather than the specific recognition of the 3-O-sulfation.

The suggestion to overexpress one HS3st (HS3st5 being the best candidate) to analyze its effect on ExtHD internalization in CHO cells, is interesting but deserves a full study (Karlsson et al., Dissecting structure-function of 3-O-sulfated heparin and engineered heparan sulfates. *Sci Adv.* 2021 Dec 24;7(52):eabl6026. doi: 10.1126/sciadv.abl6026).

The fact that the charge density is probably more relevant than the O-sulfation position itself is supported by the new internalization assays we have performed with K1 cells in the presence of soluble CS-E we have now included in the revised manuscript (Fig. S16). Indeed, as observed with soluble heparin, soluble CS-E also exhibits bimodal concentration-dependent effects at 37°C, promoting the internalization of ExtHD at low GAG concentration (in a lesser extent than with heparin) and strongly inhibiting internalization at stoichiometric amounts. These new data suggest a potential role of cell membrane CS-E in En2 internalization and confirm the charge density as a crucial parameter for En2 internalization.

The following sentences have been added p. 15 in the discussion of the revised manuscript:

In addition, the presence of soluble CS-E in the extracellular medium was shown to either promote or inhibit En2 internalization in K1 cells, depending on the GAG concentration, as observed with soluble heparin (Fig. S16). These data suggest that membrane surface CS-E could also play a role in En2 internalization.

6- Page 12: “indicating that N-sulfation of the amino sugar is not mandatory for the binding to ExtHD”. The fact that GalNAc containing CS-E and heparin oligosaccharides binds similarly to ExtHD does not necessarily mean to me that GlcNS residues of heparin are not involved. Spatial distribution of sulfates may be different in HS and CS.

We agree with the reviewer and have removed this comment from the manuscript.

7- Page 13: the authors claims that affinity of ExtHD (and HD) is greater for Heparin than for CS-E. I would be more cautious on this conclusion. Whereas Heparin is mostly composed of IdoA2S, GlcNS6S disaccharide, the proportion of CS-E units in a CS-E polysaccharide can vary a lot according to the polysaccharide origin. A disaccharide analysis of both heparin and CS-E would be needed to clarify this. Besides charge content, distribution of E units along the chain (in blocks or dispersed) may also be critical. It is therefore difficult to know whether the CS-E used in this study is structurally representative of that found in a relevant tissue/cell surface (or PNN). In line with this, analysis of the interaction of ExtHD with synthetic CS-E dp6 (exclusively composed of E units) by NMR show fairly similar binding (figure S9 and table S2). This should be discussed in the manuscript.

We thank the reviewer for their relevant comment. It would be indeed very interesting to analyze and characterize the disaccharide and oligosaccharide motif(s) that are recognized at the cell-surface GAGs by the protein. This is clearly a challenged and full study that could not be explored in this study.

The conclusion about differences in affinity is based on ITC data ($\Delta G/n$) and NMR data with dp6 oligosaccharides. We agree that the distribution of CS-E in blocks or dispersed may be critical. This is why we chose a synthetic dp6 oligosaccharide exhibiting the highest density of CS-E to represent a block distribution of CS-E. The NMR study of the interaction between ExtHD and the chemically synthesized dp6 CS-E versus the dp6 heparin fragment clearly indicates at the residue level a better affinity of the protein for dp6 heparin, with 2- to 3-fold differences in K_d^{app} at the residue level, in particular in the GAG-binding *N*-terminal motif spanning residues 192-204, as shown in Table S2.

In addition, characterizing the structure of GAG binding partners of ExtHD at the cell-surface is only a first step in the understanding of the link between binding and internalization. In fact, evidence for a binding partner is not evidence for its implication in the internalization process as we previously demonstrated for sialic acids and cell-penetrating peptides (CPPs). In this latter case, sialic acids binding to CPPs prevent the internalization of the peptides in cells (Relationships between membrane binding, affinity and cell internalization efficacy of a cell-penetrating peptide: penetratin as a case study. *PLoS One*. 2011;6(9):e24096. doi: 10.1371/journal.pone.0024096 ; Molecular partners for interaction and cell internalization of cell-penetrating peptides: how identical are they? *Nanomedicine (Lond)*. 2012 Jan;7(1):133-43. doi: 10.2217/nnm.11.165).

Finally, it has recently been shown that variation in the molecular composition and glycan structure of CSPGs contribute to the heterogeneity of PNNs (Miyata et al. (2018) Structural Variation of Chondroitin Sulfate Chains Contributes to the Molecular Heterogeneity of Perineuronal Nets. *Front. Integr. Neurosci*. 12:3. doi: 10.3389/fnint.2018.00003). This heterogeneity of PNNs is crucial since it impacts the localization of the homeoprotein Otx2, thus controls the functional maturation of PV cells. It is thus very difficult to define a single CS-E structure representative of that found in a relevant tissue/cell surface (or PNN). We have now included this information page 17 in the discussion part of the manuscript.

8- I find the dual role of HS in the internalization of En2 through endocytosis and direct translocation across the plasma membrane very interesting. Surprisingly, the authors chose to depict only the translocation mechanism in their figure 5 model. I understand that this process is their main focus, but I feel that the figure should gain by including both internalization mechanisms. Clatrin dependent HSPG mediated endocytosis has been reported in previous studies. Did the authors try to inhibit this pathway in their 37°C internalization assay?

We thank the reviewer for their relevant comment. We did not try to inhibit the clathrin-dependent pathway since the use of endocytosis inhibitors can lead to biased results (Vercauteren et al. The use of inhibitors to study endocytic pathways of gene carriers: optimization and pitfalls. *Mol Ther.* (2010) Mar;18(3):561-9. doi: 10.1038/mt.2009.281). We do agree that our data might also indicate a role of HS in the internalization of En2 through endocytosis and have now added this internalization pathway in the model drawn in Figure 5.

9- In figure 2A-B, the authors show the role of HS in ExtHD internalization using CHO mutant cells or heparinase/chondroitinase digestions. However, these experiments do not provide any information about a potential role of CS-E. As stated by the authors, CHO cells do not produce CS-E. To address this, the internalization assay would have to be performed on CS-E producing cells. Alternatively, could soluble CS-E be used in competition, as in figure 2C-D? It would also be interesting to know whether CS-E could mediate both endocytosis and direct translocation mechanisms.

We thank the reviewer for their interesting comments. As mentioned above, we have performed new internalization experiments with K1 cells in the presence of soluble CS-E (40 disaccharides, \approx 50 sulfates/mol) or heparin dp20 (10 heparin-derived disaccharides, \approx 25 sulfates/mol) (Fig. S16). As observed with soluble heparin (20 disaccharides, \approx 50 sulfates/mol), soluble CS-E also exhibits bimodal concentration-dependent effects at 37°C, promoting the internalization of ExtHD at low GAG concentration (in a lesser extent than with heparin) and strongly inhibiting internalization at stoichiometric amounts. These additional data suggest that highly-sulfated CS-E may also play a role in En2 internalization, which is consistent with the charge-based binding mode of our ExtHD protein. We have added the following sentences in the discussion of the revised manuscript:

In addition, the presence of soluble CS-E in the extracellular medium was shown to either promote or inhibit En2 internalization in K1 cells, depending on the GAG concentration, as observed with soluble heparin (Fig. S16). These data suggest that membrane surface CS-E could also play a role in En2 internalization.

It should be noted that we have also performed new internalization experiments with SKOV3 cancer cells that are known to be enriched in HS and CS-E GAGs (Hillemeier, L. et al. *Int. J. Mol. Sci.* (2022), 23, 5793. DOI:10.3390/ijms23105793 ; Gerdy et al. *Am J Pathol* (2007), 171:1324 –1333; DOI:10.2353/ajpath.2007.070111; Vallen et al. *PLoS ONE* (2014) 9(11): e111806. DOI:10.1371/journal.pone.0111806). In these SKOV3 cells, internalization of ExtHD is not significantly higher than in K1 cells (see internalization results below). Interestingly, internalization in CHO-K1 is more sensitive to heparinases I-III treatment than in SKOV3 cells, which might suggest a role of cell-membrane CS-E in ExtHD internalization. However, we could not confirm this assumption as all our attempts failed to digest/remove CS from SKOV3 cell membrane with chondroitinases ABC. For this reason, we did not include these additional experiments with SKOV3 cells in the revised manuscript.

[REDACTED]

Regarding the internalization mechanism, it is interesting to note that the presence of soluble heparin dp20 only resulted in the inhibition of ExtHD internalization at 37°C, regardless of the oligosaccharide concentration (Fig. S16), and the binding of dp20 to ExtHD did not induce formation of aggregated species (Fig. S4). This indicates that only long and highly sulfated GAGs chains (> dp20) can promote En2 endocytosis through the formation of high molecular weight aggregates. The following sentence has been added page 7 in the results section:

In contrast, we did not observe aggregated species with the heparin fragment dp20 (Fig. S4) and concomitantly did not measure any increase in ExtHD internalization in the presence of dp20, in contrast to heparin (Fig. S16).

10- Table 1: I do not understand the 80- charges for 12 kDa Heparin. As stated by the authors, 12 kDa Heparin is ~20 disaccharide, and the average charge of heparin is ~2.5-2.8 charges per disaccharide (with variations according to the source of material), which makes an average 50-56 global negative charge. Again, disaccharide analysis of the heparin used would clarify this.

We sincerely thank the reviewer for their relevant comment. The average negative charge of 12 kDa heparin is indeed about 55. We have corrected the value in Table 1 accordingly.

Minor issues

1- Figure 1 : HI not defined. Heparin ? Why HI?

We have used HI as an abbreviation for heparin to distinguish from HS that generally stands for heparan sulfates. We have now defined HI in Figure 1.

2- The authors used a very elegant MS-based approach to quantify protein internalization. Although the method has been thoroughly described elsewhere, addition of a supplementary figure summarizing the principle of the technique may be considered ?

We thank the reviewer for their comment. We have now added a supplementary figure (Fig. S12) to describe the principle and main steps of the experiment.

3- Table 3 : I presume Kd are in nM, not mM

The unit mM indicated in Table 3 is correct. As stated in the figure legend, since heparin and CS-E are different in terms of disaccharide length, Kd measured by ITC represent the additional binding affinity of each peptide molecule along the polymer. We have therefore divided the measured free energy value by the number of proteins (stoichiometry, n) bound to the polymer. The calculated Kd value thus reflects the mean value of binding affinity of one protein by polymer, being either heparin or CS-E. This is already explained in the supplemental information page S10 in the paragraph "Analysis of the thermodynamic parameters of the interactions of En2 proteins with the CS-E GAG type by ITC".

4- page 23 BSA : bovine and not bovin

We have now corrected this typo.

Reviewer #2 (Remarks to the Author):

In this manuscript the authors discovered that Engrailed-2 (En2) is a novel HS-binding protein and performed a series of biochemical and biophysical experiments to characterize this En2-HS interaction. While the authors did a decent job in characterizing the interaction, their study is incomplete for three reasons as will be detailed below (points 1-3). The biggest problem of this study however is that it is out of biological context. It is paramountly important to use a relevant cell type to investigate the role of HS in En2 internalization, which might be quite different from what they observed in CHO-K1 cells.

We thank the reviewer for their analysis and comment. We do agree that CHO cells are not the cells that are relevant for investigating En2 function in humans or other animals. However, working with cultured neuronal cells from the central nervous system (CNS) could raise more questions than answers. Dissected central nervous system cells are indeed very heterogenous in terms of cell types (notably in the cortex) since cells obtained from dissected regions are all a mixture of neurons, interneurons, and glial cells. Even in brain structures that are considered as less heterogenous regions, such as the striatum, there is still a question of cell heterogeneity. Thus, working with cultured neuronal cells would have implied working with different cells types and GAG content. In addition, the inter-individual variability between animals would have added even more heterogeneity. Another point to take into consideration is that in the whole brain, the GAG content of cells is very different from one region to the other. CS is the major component of the extracellular matrix of the CNS, representing ~20% of its total volume (Trends Neurosci. 21(5), 207–215 (1998). doi:10.1016/S0166-2236(98)01261-2). In the CNS, the ratio of CS vs HS is 9:1, but it becomes 7:3 in the perineuronal nets (PNNs) (J. Biol. Chem. 281(26), 17789–17800 (2006). doi:10.1074/jbc.M600544200).

Thus, although the GAG content and quantity are not fully characterized yet, CHO-K1, CHOpgsA-745 or SKOV3 cells present the huge benefit of being homogenous cell lines in

terms of cell phenotype and GAG content. We have added this information p. 4 of the revised manuscript. We took these model cells as such. Our objectives are to provide molecular insights into the role of GAGs in the internalization mechanism of homeoproteins. We demonstrate herein that the protein interaction with GAGs is mostly driven by interaction with HS and that the more charge density the better the interaction and internalization of the protein.

1. Based on the NMR titration experiments using dp8 oligosaccharide, the authors concluded En2 contains two GAG-binding sites. I'm not convinced because this experiment used dp8, much shorter than the required minimum binding length (dp12) as suggested by the microarray analysis. It is quite possible that these two sites work in a synergistic manner when exposed to dp12. The evidence presented in Fig.S8 is not sufficient to exclude this possibility. A far better way to reveal whether the two motifs really work together to form a complete HS-binding site would be to truncate/mutate a few basic residues at the C-terminal sequence.

We agree that both the microarray analysis and ITC indicated that our ExtHD protein can accommodate heparin fragments containing ~6 disaccharides (dp12). We have thus recorded additional NMR titration experiments with heparin dp12 to exclude the possibility that the two GAG-binding sites of En2 may bind this longer oligosaccharide in a synergetic manner. Despite the lower stability of the NMR samples in the presence of heparin dp12, we could measure chemical shift perturbations (CSP) and estimate apparent thermodynamic affinities (K_d^{app}) at the residue level. As shown in the Fig. S9, the CSP profile obtained with heparin dp12 is very similar to the ones obtained with heparin dp8 and dp6, confirming the presence of two GAG-binding sites, respectively in the N- and C-terminal regions of the protein.

The highest binding affinities for heparin dp12 are observed in the N-terminal region 194-204 while the C-terminal region 256-258 binds heparin dp12 with a significantly lower affinity (Fig. 3 and Table S2). Furthermore, truncation of the N-terminal GAG-binding site in our En2 HD protein did not modify the perturbation profile of the homeodomain region containing the C-terminal GAG-binding site (Fig. S9), as observed with heparin dp8. Taken together, these data confirm the existence of two independent interacting sites that bind heparin oligosaccharides from dp6 to dp12 with differential affinities.

In the revised manuscript, we have replaced the NMR titration experiment recorded with heparin dp8 (Fig. 3) by that recorded with heparin dp12. The corresponding analysis has been modified accordingly. It should be noted, however, that the lower stability of the NMR samples did not allow the structural characterization of the interaction with heparin dp12 as we did with heparin dp8 in Figure 4. Chemical shift deviations (CSD) and $\{^1\text{H}\}$ - ^{15}N NOEs obtained in the presence of heparin dp8 have thus been kept in the revised manuscript to obtain information on the structure and dynamics of ExtHD upon binding to GAGs. The following sentences have been added page 11 of the result section:

The stability of the NMR samples containing stoichiometric amounts of heparin dp12 was not long enough to record these experiments. Therefore, we used the shorter heparin dp8 fragment, which provided more stable samples and binds ExtHD with a similar CSP profile (Fig. S9) albeit with a slightly weaker affinity (Table S2).

Finally, it should be noted that ITC data obtained with the full-length protein (new data added), N-terminal peptide, HD and ExtHD (Table 1, Fig. S5 and S6) are in total agreement with the NMR analyses. The stoichiometry values obtained from the thermodynamics analyses indeed show that the full-length protein and ExtHD both bind 0.33 heparin polymer.

On their side, the *N*-terminal peptide binds 0.14 heparin polymer and HD, 0.20 heparin polymer; the sum of these two latter values being the one obtained for ExtHD or the full-length protein (0.14 + 0.20, that is 0.34). These results clearly demonstrate that the GAG binding sites of ExtHD interact with heparin independently from each other and not synergistically.

2. There is a high probability that HS might induce EN-2 to form oligomers, which the authors did not investigate. This experiment will need to be done with dp12 to prevent aggregation as suggested by Fig. S4.

This is an interesting point raised by the reviewer. Dimerization of some Hox transcription factors homeodomains has been reported upon DNA binding. For instance, the drosophila Scr can form dimer upon DNA binding while its closest relative Antp does not (Papadopoulos et al. Dimer formation via the homeodomain is required for function and specificity of Sex combs reduced in Drosophila. *Developmental Biology* (2012) 367, 78–89). Interestingly, it has been shown for the En2 unconventional secretion mechanism (the opposite transport of the protein compared to translocation) that binding of the protein to PIP2 does not promote its oligomerization, unlike FGF2 (Di Nardo et al., *Sci. Adv.* 2020; 6 : eabc6374).

On our side, we have indeed done DLS experiments to show that large aggregates could form in the presence of heparin used at 10 $\mu\text{g/ml}$ (protein/GAG molar ratio ≈ 8) but not at 100 $\mu\text{g/ml}$ (protein/GAG molar ratio ≈ 0.8). Additional DLS experiments have been performed with the shorter heparin dp20 fragment and did not indicate the presence of large aggregates at high protein/GAG molar ratio (see new Fig. S4). Notably, the size distribution of the ExtHD protein is only slightly increased in the presence of heparin dp20 (2.9 ± 0.1 nm when free in solution versus 3.6 ± 1.2 nm in the presence of heparin dp20), suggesting that the binding does not promote protein oligomerization.

This conclusion is supported by our NMR study, as the addition of heparin fragments (dp4, dp6, dp8, or dp12) did not result in larger line widths of the NMR cross-peaks, except for some residues that are directly involved in the interaction (see for instance Fig. 3A for the interaction with heparin dp12). This clearly indicates that our ExtHD protein, and in particular the globular homeodomain, remains mostly monomeric in the presence of heparin fragments from dp4 to dp12. Altogether, our DLS and NMR experiments indicate that the binding to heparin fragments does not induce oligomerization of our ExtHD protein, even with dp12 and dp20 which size is equal or higher than the required minimum binding length indicated by the microarray analysis and ITC. The following sentences have been added in the revised manuscript:

Page 7: “In contrast, we did not observe aggregated species with the heparin fragment dp20 (Fig. S4) and concomitantly did not measure any increase in ExtHD internalization in the presence of dp20, in contrast to heparin (Fig. S16).”

Page 10: “An increase in the line width of the NMR cross-peaks was only observed for a few residues, indicating that the binding to the heparin fragment does not induce oligomerization of the globular homeodomain.”

Page 17: “It is interesting to note that our DLS and NMR data do not support the oligomerization of ExtHD when bound to heparin fragments from dp4 to dp20. Therefore, En2 oligomerization does not seem to be a prerequisite for membrane translocation, as already proposed for secretion⁵³.”

3. I don't understand why only truncated En-2 was used in the study. Does full-length En-2 bind HS? If so, how does the affinity compare to ExtHD?

We thank the reviewer for their relevant comment. We first started the study with the full-length En2 protein together with the two shorter constructs ExtHD and HD. The full-length protein indeed interacts with heparin. The affinity of En2 for heparin is lower than that of ExtHD (Table 1), but the stoichiometry is identical for the two proteins (3 proteins bound per HI polymer). This clearly indicates that there is no additional heparin binding site in the full-length protein. The lower binding affinity of full length En2 for heparin may be due to the presence of a motif enriched in anionic residues in the disordered N-terminal region. These results have now been added in the text page 8, in Table 1, and in Fig. S6.

The reason why we have not included other results with the full-length protein herein is that we encountered many difficulties in obtaining the biotin labeled protein in a yield sufficiently high (> 50%) to be used for quantification assays. We have tried many different methods to attain such yield but all were unsuccessful. In addition, we could not recover the intact protein after incubation with cells, likely because the large N-terminal disordered region is unstable and partially subject to in-cell proteolysis. This information has been added page 21 in the Methods section of the revised manuscript.

4. Another major concern is that how did the authors verify that the E.coli expressed En-2 is biologically active?

This is indeed an important point raised by the reviewer. Most biological functions of En2 in cell signaling have been established using the bacterial recombinant protein in various in vitro experiments. In particular, injection of E.coli-expressed En2 was shown to lead to axonal guidance of retinal neurons through its ability to internalize growth cones, to regulate the translation of several mitochondrial mRNAs, and to induce ATP synthesis and secretion in retinal neurons (Brunet et al., 2005, Nature, 438:94-98; Steller et al., 2012, Development, 139:215-224). Furthermore, cell internalization of E.coli-expressed En2 was reported to protect dopaminergic neurons in several mouse models of Parkinson's disease by inducing the translation of certain key nuclear-encoded subunits of mitochondrial complex I (Alvarez-Fischer et al., 2011, Nature Neuroscience, 14:1260-1266). The injection of E.coli-expressed En2 was also shown to protect dopaminergic neurons against apoptosis under oxidative stress (Rekaik et al., 2015, Cell Reports, 13:242-250).

From the structural point of view, En2 contains a single 60-residue globular domain, the C-terminal homeodomain, and a large N-terminal intrinsically disordered region of 200 residues. The ¹H,¹⁵N chemical shifts of the homeodomain region in the E.coli-expressed ExtHD (Fig. 3A) and HD (Fig. S9) proteins are very similar to those previously reported for the native helix-turn-helix fold of En2 homeodomain (Carlier et al., 2013, Biophysical J, 105:667-678). This indicates that the region of the homeodomain adopts its native fold in our E.coli-expressed proteins. This information has now been added page 20 in the Methods section.

Reviewer #3 (Remarks to the Author):

The manuscript investigates the mechanism of the cellular internalization of the Engrailed-2 (En2), a "mobile" transcription factor that is involved in the brain development of vertebrates, focusing on the crucial role of GAG at the level of the cell membrane. The authors hypothesize that the interaction with highly sulfated GAG guides the internalization of the protein En2. To demonstrate this hypothesis, a multimodal and complementary analysis, based on MALDI mass spectrometry measurements, bindings thermodynamic and

conformational changes study, was performed to study the biochemistry of the cellular internalization of the protein. The authors could show how chemically the membrane composition can affect the internalization of the transcription factor En2 that was not well understood previously. The understanding of biochemistry beyond protein internalization is highly interesting. Significant results, obtained successfully by MALDI MS, show that protein internalization yields are driven by the presence of an N-terminal basic extension and different sulfate patterns on the cell membrane. The topic will be relevant to other fields such as oncology since the En2 is involved also in cancerogenesis. I found the manuscript well-structured and written as a non-native English speaker. The multi-modal approach employed by the authors supports the conclusions of the mechanism of internalization of En2.

For this reason, I recommend the publication after addressing my concerns about the introduction, the MALDI MS part of material and methods and supporting information.

I found the introduction well written but I think it is missing the reason why it is important to study the internalization of En2 and why it is important to understand the mechanism to emphasize the importance of the work especially for a Nature journal. I have found several publications regarding the role of En2 not only in neural formation but also in cancer such as prostate cancer (<https://www.nature.com/articles/s41598-019-41678-0>) I think giving other examples of applications can catch the interests of researchers, who are working in different fields.

As an expert in MALDI-TOF MS, I found a very innovative method to study the cellular uptake of proteins especially because they can give quantitative data of the internalized protein. However, I have the following concern about the method:

The authors mentioned that “The validity of such an approach originates from the properties of both proteins: same sequence implying identical ionization yield, similar incorporation in the MALDI matrix, same peak broadening due to attachment of alkaline and matrix adducts, identical detection efficiency due to the small mass difference”. In my experience, differences in ionization efficiency can occur when spotting the same analyte on the target plate is performed manually. How is the sample spotted on the target plate to ensure that the sampling and therefore the incorporation of the protein with the MALDI matrix are the same? I think the authors should specify the matrix application in the material methods and prove the technical reproducibility. The MALDI-MS was performed three times. Is it possible to estimate the coefficient of variation to show the technical reproducibility of the method especially the spot application on the target plate?

We thank the reviewer for their comment. We have a long experience of MALDI-TOF MS analysis of proteins and peptides. Spotting the samples manually on the MALDI plate has never proved to give unreproducible results. Identical results of quantification were obtained from spotting twice the same sample. The quantification variation that is shown for each protein comes from the independent experiments that were reproduced, thus from the biological variations, not the MS analysis.

It should be mentioned that the statistical aspect of the quantitation is important such as the sample surface area under the laser focused spot (typically, 90% of the sample area are irradiated) and the total number of laser shots (> ~10 000) allowing the detection of a large number of ions. Different mass spectra on a same sample show that the peak area of the non-labeled and labeled peak are in good agreement ensuring that the statistic is correct.

The formulation of the matrix solution should be provided (solvent and concentration) in material and methods

We used purified CHCA from LaserBio Labs (Sofia Antipolis, France). In all MALDI-TOF experiments, we worked with saturated CHCA solution in 1:1 ACN:H₂O (0.1% TFA). This information has now been added page 22 in the Methods section of the manuscript.

Which calibrant was employed for the MALDI MS measurement?

We have used classical calibrant mixture from LaserBioLabs (Sophia Antipolis, France) such as Peptide Mix3 and ProteinMix2. This information has now been added page 22 in the protocol section of the manuscript. These calibrants can cover the m/z range of peptides (the singly and doubled charged) for a relevant calibration. Calibrant sample and analyte sample were deposited side by side on the sample holder.

Why the matrix CHCA was chosen for this study? Generally, proteins are well detectable with sinapinic acid as a matrix.

The numerous experiments we have done in MALDI-TOF based internalization assays for more than twenty years, have shown that the most suitable matrix is CHCA, in terms of sensitivity, reproducibility and spectral qualities (S/N ratio). A key point in these experiments is the release of biotinylated proteins in the matrix solution after streptavidin capture. We found the elution step more efficient using CHCA than sinapinic acid.

In the Methods section, we have now added page 22 the following sentence: "CHCA was found the most suitable matrix for elution, incorporation in the matrix crystals, and MALDI-TOF analysis."

The MALDI-TOF analysis of the mixture of the pure non-labeled and labeled proteins (from solution) also shows the same ratio of peak area as the ratio concentration indicating that there is no discrimination or bias in the MALDI experiment.

In Figure S5, the x-axis shows injection time and not mole ratio. Is there any reason for this inconsistency?

We thank the reviewer for pointing out. We now showed the correct x-axis labeling in Fig. S5.

Figure S12 is very difficult to understand. The first picture is the double charged form of the protein. The second is the single charged form and the third is the comparison between simulated and experimental mixtures. First question: I do not understand why the mass of the double charged is the same for the two isotopes? Moreover, what does mean shift at 128? This picture needs to be more described with more details.

The legend of the Figure S14 (former Fig. S12) has been changed as indicated below for a better understanding:

Fig. S14. The top and middle panels show the overlapped partial mass spectra of individual ¹⁴N ExtHD and ¹⁵N ExtHD samples. The two proteins are separated by $\Delta m = 128$ (mean labeling of the protein). On top, the doubly charged ion of ¹⁵N ExtHD, and that of ¹⁴N ExtHD after m/z shift of 128/2. In the middle, the singly charge of ¹⁵N ExtHD (blue), and ¹⁴N ExtHD (red) as well as that of the ¹⁴N ExtHD (black) after m/z shift of 128. The panel at the bottom shows the fit of the ratio $r = [^{14}\text{N}]/[^{15}\text{N}]$ for ExtHD protein using the peak of ¹⁴N and ¹⁵N separately recorded in the same conditions and mixed to form a ratio r and then compared to the experimental data of the mixture (red). The best adjustment for the MH⁺ peak is obtained for a ratio $r = 0.75$ (blue). It should be noted that satellites peaks of the protein of interest can also be simulated and indicate a very close ratio r.

In Figure S13, why the relative internalized quantity of K1 is not the same as the K1 control in Figure 2D?

We thank the reviewer for their comment. In all our studies, every time we wish to test a new experimental condition, we add the control experience to be able to compare results of experiments that have been run in parallel. We ran thus the 4°C experiments both in CHO-K1 and pgsA-745 in parallel together with the experiment in CHO-K1 cells at 37°C. The results obtained is not identical but very similar, which is often the case for quantitative analyses since there are many possibilities of variations between experiments.

I would be happy to review a revised manuscript

Reviewer #4 (Remarks to the Author):

This is a fantastic piece of work, dealing with an important topic and solved using a multidisciplinary approach, with a variety of structural and biophysical techniques on top of the molecular biology protocols employed to prepare the target protein. Technically, the work is excellent, the NMR approach is perfect, the quality of the spectra is very high and the discussion and conclusions are clearly stated. Nothing else to say.

We thank the reviewer for their kind comment. We greatly appreciate.

REVIEWER COMMENTS

Reviewer #1 (Remarks to the Author):

Dear Editor,

Please, find my conclusions regarding the revised version of the manuscript from Cardon et al. (ref NCOMMS-21-44879A), entitled "A cationic motif in Engrailed-2 homeoprotein controls cell internalization through selective interaction with heparan sulfates".

In the accompanying letter, the authors provide a thorough, point by point, argument that has met my main concerns. I would therefore like to thank the authors for taking my comments into account and hope these contributed to improve the manuscript. I concede to the authors that some of the suggested experiments would require a significant effort, and I estimate that absence of these additional data does not affect the quality of the overall study. As explained in my original assessment, I consider that this work is of high interest to the field of GAGs and I am happy to recommend it for publication in Nature communications.

Reviewer #2 (Remarks to the Author):

In this revision the authors have addressed some of my previous concerns. However, they failed to address my most important concerns regarding the biologically relevant cell type and the use of truncated form of EN-2.

The authors stated the difficulty of performing the internalization experiments using primary neurons. But the experiment can be done with commercially available neuron cell lines. Without this experiment, I'm not convinced about the role of HS in EN-2 internalization. Different cells express widely different cell surface proteins and their HS structures also vary widely, therefore it is paramountly important to use relevant cells in this type of cell biology assays.

I feel strongly that the authors will need to present data on internalization assay using full-length EN-2. The authors stated the difficulty of obtaining biotinylated FL EN-2 in high yield. But I don't understand why this would pose a problem. If the yield is low, simply start with more protein for biotinylation. Also, with regard to the difficulty of recovering intact protein, the authors can try to put the biotin tag to the C-terminus of FL EN-2 instead, which would allow detection even when the N-terminal region is degraded during internalization.

Performing the internalization experiment using the full length EN-2 is all the more important in light of the new data, where the authors showed that the affinity of the FL protein to heparin is almost 5 times weaker compared to the ExtHD domain. To me this data is very concerning because it implies that the full length protein and the ExtHD domain interact with HS in somewhat different ways. For this reason the internalization assay has to be confirmed with full length EN-2. The last thing we want is to artificially make a protein (in this case using a small fraction of the full length) bind HS more strongly, which would result in misinformation.

Reviewer #3 (Remarks to the Author):

The authors have accurately addressed all of my concerns. Therefore, I strongly recommend publication in Nature Communications.

Point-by-point response to the reviewers' comments

REVIEWER COMMENTS

Reviewer #1 (Remarks to the Author):

Dear Editor,

Please, find my conclusions regarding the revised version of the manuscript from Cardon et al. (ref NCOMMS-21-44879A), entitled "A cationic motif in Engrailed-2 homeoprotein controls cell internalization through selective interaction with heparan sulfates".

In the accompanying letter, the authors provide a thorough, point by point, argument that has met my main concerns. I would therefore like to thank the authors for taking my comments into account and hope these contributed to improve the manuscript. I concede to the authors that some of the suggested experiments would require a significant effort, and I estimate that absence of these additional data does not affect the quality of the overall study. As explained in my original assessment, I consider that this work is of high interest to the field of GAGs and I am happy to recommend it for publication in Nature communications.

AUTH.: We thank the reviewer for their comments.

Reviewer #2 (Remarks to the Author):

In this revision the authors have addressed some of my previous concerns. However, they failed to address my most important concerns regarding the biologically relevant cell type and the use of truncated form of EN-2.

The authors stated the difficulty of performing the internalization experiments using primary neurons. But the experiment can be done with commercially available neuron cell lines. Without this experiment, I'm not convinced about the role of HS in EN-2 internalization. Different cells express widely different cell surface proteins and their HS structures also vary widely, therefore it is paramountly important to use relevant cells in this type of cell biology assays.

AUTH.: We thank the reviewer for their recommendation. We have performed additional internalization experiments with Ext-HD and HD proteins in SH-SY5Y neuronal cells. We chose this cell line because it is known to contain similar HS and CS glycosaminoglycans content (doi:10.1371/journal.pone.0116641), and because these cells are widely used in the field of neurosciences, developmental biology, and glycobiology that are relevant to our present study:

https://doi.org/10.1007/978-1-0716-1398-6_60

<https://doi.org/10.1016/j.neuro.2022.07.008>

<https://doi.org/10.1093/glycob/cwab126>

[https://onlinelibrary.wiley.com/doi/10.1002/\(SICI\)1097-4547\(19961201\)46:5%3C565::AID-JNR5%3E3.0.CO;2-H](https://onlinelibrary.wiley.com/doi/10.1002/(SICI)1097-4547(19961201)46:5%3C565::AID-JNR5%3E3.0.CO;2-H)

Quantification results are shown in the Figure below (that is now Fig. S16). The internalization efficacy of ExtHD into these neuronal cells is similar than into CHO-K1 cells, while HD internalizes about two-fold better in SH-SY55 compared to CHO-K1 cells. Most interestingly, the same discriminative pattern is measured between the two proteins in both cell lines. As in CHO-K1, ExtHD internalizes indeed 4-fold better than HD in SH-SY5Y cell line, confirming the importance of the cationic motif upstream the homeodomain for En2 internalization. In addition, removing heparan sulfates (heparinases I-III) at the surface of SH-SY55 cells, reduces ExtHD internalization.

I feel strongly that the authors will need to present data on internalization assay using full-length EN-2. The authors stated the difficulty of obtaining biotinylated FL EN-2 in high yield. But I don't understand why this would pose a problem. If the yield is low, simply start with more protein for biotinylation. Also, with regard to the difficulty of recovering intact protein, the authors can try to put the biotin tag to the C-terminus of FL EN-2 instead, which would allow detection even when the N-terminal region is degraded during internalization. Performing the internalization experiment using the full length EN-2 is all the more important in light of the new data, where the authors showed that the affinity of the FL protein to heparin is almost 5 times weaker compared to the ExtHD domain. To me this data is very concerning because it implies that the full-length protein and the ExtHD domain interact with HS in somewhat different ways. For this reason, the internalization assay has to be confirmed with full length EN-2. The last thing we want is to artificially make a protein (in this case using a small fraction of the full length) bind HS more strongly, which would result in misinformation.

AUTH.: We thank the reviewer for their comment. Obtaining internalization data with the full-length En2 was actually our initial goal in this study and we spent about two years trying to label the full-length protein with biotin or to apply our MS-based quantification assay. We have to underline that we started with large amounts of protein to label it with maleimide. We have engineered three different protein constructs (this information now stands in the manuscript in the Methods section) :

1. We first produced the wild-type full-length protein that naturally contains a single cysteinyl residue in position 178. We tried many different protocols to obtain a biotin-labeling yield sufficient for MS quantification: modifying the buffer, pH or even by totally unfolding the protein with strong chaotropic agents such as guanidine (up to 3M). All

attempts resulted in biotinylation yields that did not exceed 10 to 30% of the total protein, showing that the native cysteinyl residue is indeed not easily accessible to chemical modification. It is important to point that the accurate quantification of protein internalization with our MS-based assay requires a high biotinylation yield because the unlabeled protein will compete with the biotin-labeled protein for cell internalization. After the reaction with maleimide-biotin, the unlabeled and labeled species cannot be separated by HPLC, ion exchange, size-exclusion nor avidin-resin chromatography (that requires harsh elution conditions incompatible with the protein stability and additional protein sample preparation steps).

2. We introduced a cysteinyl residue at the *N*-terminal end of the full-length protein. In that case, with many different protocol trials, we could obtain a biotinylation yield of 50%. However, running internalization assays with this protein, we could not recover the intact protein after incubation with cells, likely because the large *N*-terminal disordered region is unstable and partially subject to in-cell proteolysis.
3. Finally, we introduced a cysteinyl residue at the *C*-terminal end of the full-length protein. The different trials of biotin-labeling led us to obtain the tagged protein with a yield of 70%. However, internalization assays were not more successful and we evidenced that the *C*-terminal biotin-labeled protein could not be fished, suggesting that the attached biotin in this protein construct was not accessible to streptavidin-coated magnetic beads, likely because of steric hindrance.

Although we could not apply the MS quantification protocols to those engineered En2 full-length proteins, we would like to point that the ability of *E.coli*-expressed full-length En2 to internalize into various cell lines has been previously reported by independent research groups in publications, such as:

- Brunet *et al.*, The transcription factor Engrailed-2 guides retinal axons. *Nature* **438**, 94–98 (2005). <https://doi.org/10.1038/nature04110>;
- Punia *et al.*, Membrane insertion and secretion of the Engrailed-2 (EN2) transcription factor by prostate cancer cells may induce antiviral activity in the stroma. *Sci Rep* **9**, 5138 (2019). <https://doi.org/10.1038/s41598-019-41678-0>;
- Lee *et al.* Global Analysis of Intercellular Homeodomain Protein Transfer. *Cell Rep.* 2019 Jul 16;28(3):712-722.e3. doi: 10.1016/j.celrep.2019.06.056.

To demonstrate the ability of full-length En2 to effectively internalize into CHO-K1 cells, we have now added confocal imaging of CHO-K1 cells incubated with the CF-labeled full-length protein, which clearly indicates that the protein is indeed endowed with internalization properties, although this method does not allow the quantification of cell internalization (Fig. S19).

Fig. S19. Confocal imaging of fluorescent En2 internalization in CHO-K1 cells.

Apart from internalization assays, as mentioned by Reviewer #2, ExtHD has indeed a slightly better affinity than the full-length protein for heparin. However, the enthalpy component of this interaction is better for the full-length protein (-51 kJ/mol) compared to ExtHD (-41 kJ/mol), meaning that the full-length protein interacts stronger (more non covalent interactions) with the anionic polymer than its shortened analogue. This difference of enthalpy is however compensated for the full-length protein by a higher unfavorable entropy than ExtHD, which results in the apparent lower affinity of the full-length protein for heparin. Since En2 contains an intrinsically disordered *N*-terminal domain that is shortened in the ExtHD analogue, the entropy penalty measured for interaction of the two proteins with heparin may be explained by a higher loss in the conformational flexibility and degree of freedom of the full-length protein compared to ExtHD that does not contain the disordered *N*-terminal domain. These results highlight how intricate are binding thermodynamics and the importance of accounting for affinity, enthalpy and entropy parameters for a comprehensive analysis of the interaction molecular system. This point has now been added to the manuscript (Fig. S7), through the addition of the thermodynamic graphs (left panel) of the three proteins, Full-length, ExtHD and HD with ΔG° (grey), ΔH° (blue), and $-T\Delta S^\circ$ (red), and the plot of $-T\Delta S^\circ$ versus ΔH° indicating possible enthalpy–entropy compensation.

Fig. S7. Thermodynamic parameters of heparin (HI) interaction with En2 fragments obtained from ITC.

Reviewer #3 (Remarks to the Author):

The authors have accurately addressed all of my concerns. Therefore, I strongly recommend publication in Nature Communications.

AUTH.: We thank the Reviewer for their comments.

Paris, March 15th, 2023

Point-to-point Answers to reviewers' comments

We deeply acknowledge the four reviewers for their approval of our study to be published in Nature Communications.